# New Pyrimidinone-Fused 1,4-Naphthoquinone Derivatives Inhibit the Growth of Drug Resistant Oral Bacteria

**DOI:** 10.3390/biomedicines8060160

**Published:** 2020-06-15

**Authors:** Kyungmin Kim, Daseul Kim, Hyunjin Lee, Tae Hoon Lee, Ki-Young Kim, Hakwon Kim

**Affiliations:** 1Department of Applied Chemistry, Global Center for Pharmaceutical Ingredient Materials, Kyung Hee University, Gyeonggi-do 1732, Korea; sp10101@naver.com (K.K.); 1009hjlee@naver.com (H.L.); thlee@khu.ac.kr (T.H.L.); 2Graduate School of Biotechnology, Kyung Hee University, Gyeonggi-do 1732, Korea; charybde@naver.com

**Keywords:** 1,4-napthoquinone, pyrimidinone, antibacterial activity, oral bacteria, dental caries

## Abstract

Background: Dental caries is considered to be a preventable disease, and various antimicrobial agents have been developed for the prevention of dental disease. However, many bacteria show resistance to existing agents. Methods/Principal Findings: In this study, four known 1,4-naphthoquinones and newly synthesized 10 pyrimidinone-fused 1,4-naphthoquinones, i.e. KHQ 701, 702, 711, 712, 713, 714, 715, 716, 717 and 718, were evaluated for antimicrobial activity against *Enterococcus faecalis, Enterococcus faecium, Staphylococcus aureus, Staphylococcus epidermidis, Streptococcus mutans, Streptococcus sobrinus, Porphyromonas gingivalis, Actinomyces viscosus* and *Fusobacterium nucleatum.* Pyrimidinone-fused 1,4-naphthoquinones were synthesized in good yields through a series of chemical reactions from a commercially available 1,4-dihydroxynaphthoic acid. MIC values of KHQ 711, 712, 713, 714, 715, 716, 717 and 718 were 6.25–50 μg/mL against *E. faecalis* (CCARM 5511)*,* 6.25–25 μg/mL against *E. faecium* (KACC11954) and *S. aureus* (CCARM 3506), 1.56–25 μg/mL against *S. epidermidis* (KACC 13234), 3.125–100 μg/mL against *S. mutans* (KACC16833), 1.56–100 μg/mL against *S. sobrinus* (KCTC5809) and *P. gingivalis* (KCTC 5352), 3.125–50 μg/mL against *A. viscosus* (KCTC 9146) and 3.125–12.5 μg/mL against *F. nucleatum* (KCTC 2640) with a broth microdilution assay. A disk diffusion assay with KHQ derivatives also exhibited strong susceptibility with inhibition zones of 0.96 to 1.2 cm in size against *P. gingivalis*. Among the 10 compounds evaluated, KHQ 711, 712, 713, 715, 716 and 717 demonstrated strong antimicrobial activities against the 9 types of pathogenic oral bacteria. A pyrimidin-4-one moiety comprising a phenyl group at the C2 position and a benzyl group at the N3 position appears to be essential for physiological activity. Conclusion/Significance: Pyrimidinone-fused 1,4-naphthoquinones synthesized from simple starting compounds and four known 1,4-naphthoquinones were synthesized and showed strong antibacterial activity to the 9 common oral bacteria. These results suggest that these derivatives should be prospective for the treatment of dental diseases caused by oral bacteria, including drug-resistant strains.

## 1. Introduction

Two major dental diseases in the world are dental caries and periodontal disease, both of which are caused by various bacteria in the oral cavity [1]. Dental caries is a common oral disease that usually develops the formation of plaque biofilms on tooth surfaces, and the causative agents are Gram-positive bacteria such as *Streptococcus mutans* and *Streptococcus sobrinus,* as well as some non-mutans streptococci [2,3,4]. Specific bacterial species such as *Actinomyces* spp. and *Enterococcus faecalis* contribute to tooth root caries and periodontal infection [4,5].

Although dental disease is only slowly progressive, oral bacteria can also cause infections of the head and neck in locations such as periapical abscesses, the jaw bones and fascia [6,7,8]. Therefore, the control of oral bacteria is the key to the prevention and the treatment of these oral diseases. Various antibiotics including ampicillin, chlorhexidine, erythromycin, spiramycin and vancomycin have been used to prevent dental caries, but these agents can cause unexpected side effects such as microorganism resistance, vomiting and diarrhea [6,9]. Furthermore, the use of antibiotics can promote the development of multidrug-resistant (MDR) strains of bacteria [10]. These problems have led to a search for new antibacterial substances specific to oral pathogens [11].

1,4-Naphthoquinone (1,4-NQ) is the central chemical structure of biologically active compounds such as menadione as a coagulant, juglone as a herbicide, and dichlone as a fungicide. As such, the 1,4-NQ moiety is known as an important pharmacophore that exhibits various pharmacological properties such as antibacterial, antifungal, antiviral, pesticide and anti-inflammatory activities, thus playing an important role in the development of new drugs. Among the 1,4-naphthoquinone derivatives, heterocycle-fused naphthoquinones in particular have been demonstrated to exhibit a variety of biological activities [12,13,14,15,16].

Pyrimidine derivatives, such as pyrimidin-4-one and pyrimidine-2,4-dione, have also various biological activities such as antimicrobial, antiviral and antifungal activities, and thus are considered to be a new, major component for drug discovery [17,18].

The conjugation of pyrimidinone, which exhibits antimicrobial activity, to 1,4-naphthoquinone, which exhibits antimicrobial, antifungal and antiviral activity, could lead to the development of synthetic antimicrobial drugs having better activity [19].

With the above considerations, a series of novel pyrimidinone or pyrimidindione-fused 1,4-naphthoquinones were synthesized via a pharmacophore hybridization strategy [20].

## 2. Materials and Methods

### 2.1. General Experimental Details for Synthesis of Compounds

All chemical reagents were purchased from Sigma-Aldrich, Tokyo Chemical Industry, Alfa Aesar and Acros Organics, and were used without further purification. All glassware was thoroughly dried in a convection oven. Reactions were monitored using thin-layer chromatography (TLC). Commercial TLC plates (silica gel 60 F_254_, Merck Co.) were developed and the spots were visualized under UV light at 254 or 365 nm. Silica gel column chromatography was performed with silica gel 60 (particle size 0.040−0.063 mm, Merck Co.). Extra pure-grade solvents for column chromatography were purchased through Samchun Chemicals and Duksan Chemicals. ^1^H and ^13^C NMR spectra were collected with a JEOL ECX-400 spectrometer (at 300 MHz for ^1^H NMR and 75 MHz for ^13^C NMR), and the chemical shifts were recorded with respect to tetramethylsilane, Si(CH_3_)_4_, as an internal reference or were referenced to the residual proton peaks of the deuterated solvent. Mass spectrometry (MS) spectra were recorded using a Bruker compact ESI quadrupole-TOF ultra-high-resolution liquid chromatograph/mass spectrometer.

#### 2.1.1. Preparation of 3-bromo-1, 4-dimethoxy-2-naphthoic acid (4)

Methyl 3-bromo-1, 4-dimethoxy-2-naphthoate (**3**). To a solution of methyl 1,4-dimethoxy-2-naphthoate (2, 7.7 g, 31.3 mmol) in 78 mL of *N,N*-dimethylformamide (DMF), *N*-bromosuccinimide (NBS, 34.4 mmol, 6.1 g) was slowly added at room temperature and it was stirred overnight. After completion of the reaction, it was concentrated in vacuo and treated with ethyl acetate and water. The organic layer was separated, washed with brine and dried over Na_2_SO_4_. After filtration, the crude residue was purified by a silica gel flash column chromatography (17% ethyl acetate/hexane) to give compound **3** as a white solid (10.0 g, 98%): ^1^H NMR (DMSO-d_6_, 300 MHz) δ ppm 8.13–8.09 (m, 2H), 7.74–7.71 (m, 2H), 3.93 (s, 3H), 3.91 (s, 3H), 3.91 (s, 3H); ^13^C NMR (CDCl_3_, 75 MHz) δ ppm 166.60, 150.28, 150.23, 129.55, 128.10, 127.77, 127.13, 126.42, 123.06, 122.63, 108.57, 63.73, 61.59, 52.86.

3-Bromo-1,4-dimethoxy-2-naphthoic acid (4). Methyl 3-bromo-1,4-dimethoxy-2-naphthoate (**3**, 15.7 g, 48.4 mmol) was dissolved in 50 mL of tetrahydrofuran (THF), 50 mL of methanol and 100 mL of water. Potassium hydroxide (241.7 mmol, 13.6 g) was added at room temperature and the reaction mixture was refluxed for 48 h or until all the starting material was consumed. After completion of the reaction, the mixture was cooled to room temperature. The reaction mixture was washed with a saturated aqueous solution of K_2_CO_3_ and brine. After filtration, the residue was washed with a hexane and dissolved in water. The mixture was acidified with concentrated HCl. After filtration, the residue was washed with water and hexane. Compound 4 was obtained as a white solid (12.1 g, 84%): ^1^H NMR (DMSO-d_6_, 300 MHz) δ ppm 13.75 (s, 1H), 8.13–8.06 (m, 2H), 7.73–7.67 (m, 2H), 3.92 (s, 3H), 3.90 (s, 3H); ^13^C NMR (CDCl_3_, 75 MHz) δ ppm 171.19, 150.56, 150.43, 129.78, 128.37, 127.75, 127.27, 125.58, 123.19, 122.70, 108.15, 64.08, 61.65.

#### 2.1.2. Synthesis of 5a−5i from Naphthoic Acid

3-Bromo-1,4-dimethoxy-*N*-phenyl-2-naphthamide (5a). 3-Bromo-1,4-dimethoxy-2-naphthoic acid (4, 0.50 g, 1.6 mmol) was dissolved in 8 mL of anhydrous dichloromethane (DMC) and catalytic amount of anhydrous DMF under argon. Oxalyl chloride (0.23 mL, 2.7 mmol) was added dropwise at room temperature and the reaction was left to stir for 3 h. The solution was evaporated under reduced pressure and the oily residue was dried under high vacuum for 1 hour to ensure all the oxalyl chloride was removed. Without further purification, the oily residue was dissolved in 13 mL of anhydrous DMC and 0.32 mL anhydrous pyridine under argon. Aniline (0.18 mL, 1.9 mmol) was added dropwise and the reaction mixture was stirred at room temperature overnight. After completion of the reaction, the mixture was quenched with 3 M HCl at 0 °C and extracted with DMC. The organic layer was separated, washed with brine and dried over Na_2_SO_4_. After filtration, the crude residue was purified by a silica gel flash column chromatography (20% ethyl acetate/hexane) to give compound 5a as a white solid (0.601 g, 97% yield over two steps): ^1^H NMR (DMSO-d_6_, 300 MHz) δ ppm 10.60 (s, 1H), 8.14–8.11 (m, 2H), 7.73–7.70 (m, 4H), 7.36 (t, 2H, *J* = 7.9 Hz), 7.14–7.09 (m, 1H), 3.93 (s, 6H); ^13^C NMR (DMSO-d_6_, 75 MHz) δ ppm 163.34, 149.41, 149.17, 138.92, 129.75, 128.85, 128.46, 128.24, 127.54, 127.41, 123.82, 122.92, 122.34, 119.46, 109.74, 63.43, 61.40.

*N*-Benzyl-3-bromo-1,4-dimethoxy-2-naphthamide (5b). Following the procedure described above for the preparation of 5a, compound 5b was obtained from benzylamine (0.42 mL, 3.9 mmol) as a white solid (1.2 g, 96% yield over two steps): ^1^H NMR (DMSO-d_6_, 300 MHz) δ ppm 9.06 (t, 1H, *J* = 5.8 Hz), 8.10–8.08 (m, 2H), 7.69–7.66 (m, 2H), 7.46–7.34 (m, 4H), 7.26 (t, 1H, *J* = 7.1 Hz), 4.51 (d, 2H, *J* = 5.7 Hz), 3.90 (s, 3H), 3.85 (s, 3H); ^13^C NMR (CDCl_3_, 75 MHz) δ ppm 165.75, 150.28, 150.05, 137.65, 129.32, 128.93, 128.69, 128.05, 127.99, 127.94, 127.63, 127.10, 122.91, 122.54, 109.71, 63.92, 61.51, 44.12.

3-Bromo-*N*-cinnamyl-1,4-dimethoxy-2-naphthamide (5h). Following the procedure described above for the preparation of 5a, compound 5h was obtained from cinnamylamine (0.36 g, 2.7 mmol) as a white solid (0.871 g, 90% yield over two steps): ^1^H NMR (DMSO-d_6_, 300 MHz) δ ppm 8.82 (t, 1H, *J* = 5.7 Hz), 8.11–8.09 (m, 2H), 7.70–7.67 (m, 2H), 7.43 (d, 2H, *J* = 7.9 Hz), 7.34 (t, 2H, *J* = 7.5 Hz), 7.27–7.22 (m, 1H), 6.70 (d, 1H, *J* = 15.9 Hz), 6.39–6.32 (m, 1H), 4.08 (t, 2H, *J* = 5.3 Hz), 3.92 (s, 3H), 3.90 (s, 3H); ^13^C NMR (CDCl_3_, 75 MHz) δ ppm 165.77, 150.29, 150.05, 136.55, 132.39, 129.33, 128.96, 128.60, 127.97, 127.93, 127.74, 127.13, 126.39, 124.90, 122.94, 122.56, 109.69, 63.96, 61.53, 41.95.

3-Bromo-1,4-dimethoxy-*N*-(4-methoxybenzyl)-2-naphthamide (5i). Following the procedure described above for the preparation of 5a, compound 5i was obtained from 4-methoxybenzylamine (1.0 mL, 7.7 mmol) as a white solid (2.2 g, 80% yield over two steps): ^1^H NMR (DMSO-d_6_, 300 MHz) δ ppm 8.98 (t, 1H, *J* = 5.9 Hz) 8.10–8.06 (m, 2H), 7.70–7.67 (m, 2H), 7.35 (d, 2H, *J* = 8.4 Hz), 6.92 (d, 2H, *J* = 8.6 Hz), 4.43 (d, 2H, *J* = 5.7 Hz), 3.89 (s, 3H), 3.85 (s, 3H), 3.74 (s, 3H); ^13^C NMR (CDCl_3_, 75 MHz) δ ppm 165.64, 159.08, 150.24, 150.02, 129.73, 129.39, 129.28, 128.95, 127.93, 127.88, 127.08, 122.90, 122.52, 114.04, 109.74, 63.91, 61.50, 55.28, 43.61.

#### 2.1.3. Synthesis of 6a-6i from 5a−5i

3-Amino-1,4-dimethoxy-*N*-phenyl-2-naphthamide (6a). 3-Bromo-1,4-dimethoxy-*N*-phenyl-2-naphthamide (0.54 g, 1.4 mmol), copper iodide (0.27 g, 1.4 mmol), L-proline (0.21 g, 1.8 mmol), and sodium azide (0.18 g, 2.8 mmol) were dissolved in 2.8 mL of anhydrous dimethyl sulfoxide (DMSO) under argon. The reaction mixture was stirred at 100 °C for 3 h. After completion of the reaction, the solution was cooled to room temperature and quenched by the addition of saturated aqueous NH_4_Cl and ethyl acetate. This biphasic mixture was stirred at room temperature for 1 hour. The resulting solution was filtered through a pad of celite which was subsequently washed with ethyl acetate and water. The filtrate was transferred to a separatory funnel, the aqueous phase was removed and the organic phase was washed with a saturated NaHCO_3_ solution and brine. The aqueous phases were combined and extracted with ethyl acetate. This washing/extraction procedure was repeated two additional times. Finally, the organic phases were combined, dried over Na_2_SO_4_, concentrated under reduced pressure, and purified by the silica gel flash column chromatography with 14% ethyl acetate/hexane. Compound 6a was obtained as a yellow solid (0.254 g, 57%): ^1^H NMR (DMSO-d_6_, 300 MHz) δ ppm 10.47 (s, 1H), 7.93 (d, 1H, *J* = 8.3 Hz) 7.81–7.76 (m, 3H) 7.48 (t, 1H, *J* = 7.5 Hz), 7.35 (t, 2H, *J* = 7.8 Hz), 7.26 (t, 1H, *J* = 7.5 Hz), 7.11 (t, 1H, *J* = 7.4 Hz), 5.08 (d, 2H, *J* = 5.1 Hz), 3.84 (s, 3H), 3.75 (s, 3H); ^13^C NMR (CDCl_3_, 75 MHz) δ ppm 165.14, 152.06, 137.95, 137.71, 136.51, 129.89, 129.10, 128.35, 124.60, 123.10, 122.77, 120.45, 120.37, 120.05, 113.13, 63.53, 59.68.

3-Amino-*N*-benzyl-1,4-dimethoxy-2-naphthamide (6b). Following the procedure described above for the preparation of **6a**, compound **6b** was obtained from **5b** (0.71 g, 1.8 mmol) as a yellowish brown solid (0.418 g, 70%): ^1^H NMR (DMSO-d_6_, 300 MHz) δ ppm 8.99 (t, 1H, *J* = 7.8 Hz), 7.89 (d, 1H, *J* = 8.4 Hz), 7.76 (d, 1H, *J* = 8.4 Hz), 7.47–7.33 (m, 5H), 7.28–7.21 (m, 2H), 5.11 (d, 2H, *J* = 5.5 Hz), 4.50 (d, 2H, *J* = 6.0 Hz), 3.73 (s, 3H), 3.72 (s, 3H); ^13^C NMR (CDCl_3_, 75 MHz) δ ppm 166.84, 152.14, 138.11, 137.53, 136.32, 129.71, 128.76, 128.07, 128.02, 127.56, 123.05, 122.56, 120.52, 120.00, 113.13, 63.23, 59.65, 43.81.

3-Amino-*N*-cinnamyl-1,4-dimethoxy-2-naphthamide (6h). Following the procedure described above for the preparation of 6a, compound 6h was obtained from 5h (0.65 g, 1.5 mmol) as a bright yellow solid (0.31 g, 55%): ^1^H NMR (DMSO-d_6_, 300 MHz) δ ppm 8.74 (t, 1H, *J* = 5.6 Hz), 7.91 (d, 1H, *J* = 8.4 Hz), 7.76 (d, 1H, *J* = 8.4 Hz), 7.48–7.43 (m, 3H), 7.34 (t, 2H, *J* = 7.4 Hz), 7.26–7.22 (m, 2H), 6.65 (d, 1H, *J* = 16.1 Hz), 6.43–6.34 (m, 1H), 5.13 (d, 2H, *J* = 5.7 Hz), 4.10 (t, 2H, *J* = 5.5 Hz), 3.83 (s, 3H), 3.73 (s, 3H); ^13^C NMR (CDCl_3_, 75 MHz) δ ppm 166.86, 152.11, 137.46, 136.53, 136.33, 132.48, 129.71, 128.60, 128.08, 127.75, 126.38, 125.13, 123.08, 122.60, 120.56, 120.01, 113.22, 63.24, 59.67, 41.66.

3-Amino-1,4-dimethoxy-*N*-(4-methoxybenzyl)-2-naphthamide (6i). Following the procedure described above for the preparation of 6a, compound 6i was obtained from 5i (0.65 g, 1.5 mmol) as a yellowish brown solid (0.33 g, 59%): ^1^H NMR (DMSO-d_6_, 300 MHz) δ ppm 8.92 (t, 1H, *J* = 5.9 Hz), 7.88 (d, 1H, *J* = 8.4 Hz), 7.75 (d, 1H, *J* = 8.4 Hz), 7.47–7.42 (m, 1H), 7.34 (d, 2H, *J* = 8.3 Hz), 7.23 (t, 1H, *J* = 7.6 Hz), 6.92 (d, 2H, *J* = 8.4 Hz), 5.11 (d, 2H, *J* = 5.7 Hz), 4.43 (d, 2H, *J* = 5.9 Hz), 3.74 (s, 3H), 3.72 (s, 3H), 3.70 (s, 3H); ^13^C NMR (CDCl_3_, 75 MHz) δ ppm 166.71, 159.02, 152.08, 137.49, 136.28, 130.22, 129.66, 129.37, 128.03, 123.00, 122.52, 120.53, 119.98, 114.09, 113.20, 63.20, 59.63, 55.27, 43.25.

#### 2.1.4. Synthesis of 7a or 7b from 6a or 6b

5,10-Dimethoxy-3-phenylbenzo[*g*]quinazoline-2,4(*1H*,*3H*)-dione (7a). 3-Amino-1,4-dimethoxy-*N*-phenyl-2-naphthamide (150 mg, 0.47 mmol) and triphosgene (69 mg, 0.23 mmol) were dissolved in 9.3 mL THF. The reaction mixture was refluxed for 3 h. After completion of the reaction, the mixture was cooled to room temperature and treated with ethyl acetate and water. The organic layer was separated, washed with a brine and dried over Na_2_SO_4_. After filtration, the crude residue was purified by a silica gel flash column chromatography (33% ethyl acetate/hexane) to give compound 7a as a bright yellow solid (0.139 g, 86%): ^1^H NMR (DMSO-d_6_, 300 MHz) δ ppm 11.02 (s, 1H), 8.21 (d, 1H, *J* = 8.4 Hz), 8.02 (d, 1H, *J* = 9.0 Hz), 7.74–7.69 (m, 1H), 7.55–7.42 (m, 4H), 7.34 (d, 2H, *J* = 7.9 Hz), 3.90 (s, 3H), 3.87 (s, 3H); ^13^C NMR (CDCl_3_, 75 MHz) δ ppm 160.56, 156.69, 150.35, 135.55, 134.96, 131.02, 130.03, 129.43, 128.82, 128.80, 126.66, 125.55, 125.35, 124.74, 120.94, 106.93, 63.61, 62.03.

3-Benzyl-5,10-dimethoxybenzo[*g*]quinazoline-2,4(*1H*,3H)-dione (7b). Following the procedure described above for the preparation of 7a, compound 7b was obtained from 6b (160 mg, 0.48 mmol) as a white yellow solid (0.142 g, 82%): ^1^H NMR (DMSO-d_6_, 300 MHz) δ ppm 11.02 (s, 1H), 8.21 (d, 1H, *J* = 8.4 Hz), 8.01 (d, 1H, *J* = 8.4 Hz), 7.72–7.67 (m, 1H), 7.54–7.49 (m, 1H), 7.36–7.23 (m, 5H), 5.12 (s, 2H), 3.92 (s, 3H), 3.84 (s, 3H); ^13^C NMR (CDCl_3_, 75 MHz) δ ppm 160.21, 156.26, 150.50, 137.09, 135.37, 130.81, 129.84, 128.96, 128.39, 127.50, 126.57, 125.35, 125.19, 124.58, 120.80, 106.76, 63.48, 61.96, 43.88.

#### 2.1.5. Synthesis of KHQ 701 or KHQ 702 from 7a or 7b

3-Phenylbenzo[*g*]quinazoline-2,4,5,10(*1H*,*3H*)-tetraone (KHQ 701). To a solution of 5,10-dimethoxy-3-phenylbenzo[*g*]quinazoline-2,4(*1H*,*3H*)-dione (80 mg, 0.23 mmol) in 2.0 mL acetonitrile and 2.0 mL DMF, CAN (567 mg, 1.03 mmol) in 2.0 mL of distilled water was slowly added at 0 °C for 10 min. After the addition was complete, the mixture was stirred for an additional 1 hour at room temperature. The resulting mixture was treated with ethyl acetate and water. The organic layer was separated, washed with a brine and dried over Na_2_SO_4_. After filtration, the crude residue was purified by a silica gel flash column chromatography (50% ethyl acetate/hexane, 100% ethyl acetate, 100% dichloromethane and then 20% methanol/dichloromethane) to give compound KHQ 701 as a yellow solid (0.073 g, quantitative): mp 285 °C (decomposed); ^1^H NMR (DMSO-d_6_, 300 MHz) δ ppm 12.23 (s, 1H), 8.06 (d, 2H, *J* = 7.9 Hz), 7.94–7.83 (m, 2H), 7.50–7.40 (m, 3H), 7.21 (d, 2H, *J* = 7.1 Hz); HRMS (ESI^+^) for C_18_H_10_N_2_O_4_Na (M^+^ •Na) calcd, 341.0533; found, 341.0535.

3-Benzylbenzo[*g*]quinazoline-2,4,5,10(*1H*,*3H*)-tetraone (KHQ 702). Following the procedure described above for the preparation of KHQ 701, compound KHQ 702 was obtained from 7b (0.168 g, 0.46 mmol) as a yellow solid (0.154 g, quantitative).: mp 293 °C (decomposed); ^1^H NMR (DMSO-d_6_, 300 MHz) δ ppm 12.18 (s, 1H), 8.05–8.02 (m, 2H), 7.99–7.76 (m, 2H), 7.28–7.21 (m, 5H), 5.01 (s, 2H); HRMS (ESI^+^) for C_19_H_12_N_2_O_4_Na (M^+^•Na) calcd, 355.0689; found, 355.0691.

#### 2.1.6. Synthesis of 7c−7i from 6a-6i

5,10-dimethoxy-2,3-diphenyl-2,3-dihydrobenzo[*g*]quinazolin-4(*1H*)-one (7c). 3-Amino-1,4-dimethoxy-*N*-phenyl-2-naphthamide (150 mg, 0.47 mmol) and *p*-toluenesulfonic acid monohydrate (8.9 mg, 0.047 mmol) were dissolved in 4.7 mL THF. Benzaldehyde (0.071 mL, 0.70 mmol) was added dropwise at room temperature. After stirring at 40 °C for 6 h, the resulting mixture was treated with ethyl acetate and water. The organic layer was separated, washed with a brine and dried over Na_2_SO_4_. After filtration, the mixture was concentrated under reduced pressure, and the crude residue was purified by a silica gel flash column chromatography (10% ethyl acetate/hexane) to give compound 7c as a yellow solid (0.176 g, 92%).: ^1^H NMR (DMSO-d_6_, 300 MHz) δ ppm 8.03 (d, 1H, *J* = 8.3 Hz), 7.72 (d, 1H, *J* = 8.4 Hz), 7.50–7.47 (m, 4H), 7.43–7.40 (m, 4H), 7.32–7.23 (m, 5H), 6.15 (d, NH, *J* = 3.1 Hz), 3.93 (s, 3H), 3.66 (s, 3H); ^13^C NMR (DMSO-d_6_, 75 MHz) δ ppm 160.07, 155.13, 141.26, 140.69, 135.07, 134.62, 130.37, 128.94, 128.91, 128.78, 128.42, 128.12, 126.23, 125.64, 123.80, 122.91, 122.14, 119.53, 110.19, 72.19, 62.37, 60.40.

3-Benzyl-5,10-dimethoxy-2-phenyl-2,3-dihydrobenzo[*g*]quinazolin-4(*1H*)-one (7d). Following the procedure described above for the preparation of 7c, compound 7d was obtained from 6b (80 mg, 0.24 mmol) as a yellow solid (0.093 g, 92%).: ^1^H NMR (DMSO-d_6_, 300 MHz) δ ppm 8.01 (d, 1H, *J* = 7.9 Hz), 7.68 (d, 1H, *J* = 8.4 Hz), 7.49–7.44 (m, 1H), 7.42–7.33 (m, 6H), 7.30–7.21 (m, 6H), 5.73 (d, NH, *J* = 3.3 Hz), 5.47 (d, 1H, *J* = 15.4 Hz), 4.03 (d, 1H, *J* = 15.2 Hz), 3.94 (s, 3H), 3.58 (s, 3H); ^13^C NMR (CDCl_3_, 75 MHz) δ ppm 161.14, 155.59, 139.38, 136.90, 135.16, 134.49, 130.73, 129.03, 128.96, 128.94, 128.63, 128.03, 127.46, 126.11, 124.51, 123.94, 123.26, 119.82, 109.95, 70.38, 63.27, 60.62, 47.35.

3-Benzyl-5,10-dimethoxy-2-p-tolyl-2,3-dihydrobenzo[*g*]quinazolin-4(*1H*)-one (7e). Following the procedure described above for the preparation of 7c, compound 7e was obtained from 6b (100 mg, 0.30 mmol) as a bright yellow solid (0.125 g, 96%).: ^1^H NMR (DMSO-d_6_, 300 MHz) δ ppm 8.01 (d, 1H, *J* = 8.3 Hz), 7.67 (d, 1H, *J* = 8.3 Hz), 7.47 (t, 1H, *J* = 7.5 Hz), 7.41–7.33 (m, 4H), 7.30–7.21 (m, 4H), 7.16 (d, 1H, *J* = 3.1 Hz), 7.07 (d, 2H, *J* = 8.1 Hz), 5.68 (d, NH, *J* = 2.9 Hz), 5.48 (d, 1H, *J* = 15.2 Hz), 3.99 (d, 1H, *J* = 15.4 Hz), 3.94 (s, 3H), 3.58 (s, 3H), 2.17 (s, 3H); ^13^C NMR (CDCl_3_, 75 MHz) δ ppm 161.19, 155.55, 138.94, 136.99, 136.39, 135.15, 134.63, 130.72, 129.57, 128.62, 128.59, 128.02, 127.41, 126.08, 124.51, 123.90, 123.21, 119.81, 110.01, 70.19, 63.26, 60.62, 47.27, 21.07.

3-Benzyl-2-(4-fluorophenyl)-5,10-dimethoxy-2,3-dihydrobenzo[*g*]quinazolin-4(*1H*)-one (7f). Following the procedure described above for the preparation of 7c, compound 7f was obtained as a bright yellow solid (170 mg, 99%) by reaction with 6b (0.130 g, 0.39 mmol).: ^1^H NMR (DMSO-d_6_, 300 MHz) δ ppm 8.02 (d, 1H, *J* = 8.4 Hz), 7.69 (d, 1H, *J* = 8.0 Hz), 7.48 (t, 1H, *J* = 7.4 Hz), 7.42–7.33 (m, 6H), 7.30–7.25 (m, 2H), 7.23–7.20 (m, 1H), 7.11 (t, 2H, *J* = 8.9 Hz), 5.74 (d, NH, *J* = 3.8 Hz), 5.43 (d, 1H, *J* = 15.4 Hz), 4.07 (d, 1H, *J* = 15.2 Hz), 3.94 (s, 3H), 3.58 (s, 3H); ^13^C NMR (CDCl_3_, 75 MHz) δ ppm 164.58, 161.30, 161.04, 155.64, 136.74, 135.28, 135.24, 135.22, 134.26, 130.74, 128.74, 128.65, 128.07, 127.98, 127.97, 127.50, 124.51, 123.98, 123.40, 119.84, 115.99, 115.71, 109.82, 69.85, 63.30, 60.64, 47.32.

3-Benzyl-5,10-dimethoxy-2-(4-methoxyphenyl)-2,3-dihydrobenzo[*g*]quinazolin-4(*1H*)-one (7g). Following the procedure described above for the preparation of 7c, compound 7g was obtained as a bright yellow solid (198 mg, 98%) by reaction with 6b (0.150 g, 0.45 mmol).: ^1^H NMR (DMSO-d_6_, 300 MHz) δ ppm 8.01 (d, 1H, *J* = 8.4 Hz), 7.68 (d, 1H, *J* = 8.4 Hz), 7.47 (t, 1H, *J* = 7.6 Hz), 7.41–7.33 (m, 4H), 7.29–7.24 (m, 4H), 7.12 (d, 1H, *J* = 2.8 Hz), 6.83 (d, 2H, *J* = 8.8 Hz), 5.67 (d, NH, *J* = 2.9 Hz), 5.46 (d, 1H, *J* = 15.4 Hz), 3.98 (d, 1H, *J* = 15.5 Hz), 3.94 (s, 3H), 3.64 (s, 3H), 3.58 (s, 3H); ^13^C NMR (CDCl_3_, 75 MHz) δ ppm 161.19, 160.08, 155.55, 137.02, 135.14, 134.69, 131.37, 130.71, 128.61, 128.58, 127.98, 127.57, 127.37, 124.50, 123.89, 123.23, 119.82, 114.17, 109.98, 70.12, 63.27, 60.61, 55.26, 47.17.

3-Cinnamyl-5,10-dimethoxy-2-phenyl-2,3-dihydrobenzo[*g*]quinazolin-4(*1H*)-one (7h). Following the procedure described above for the preparation of 7c, compound 7h was obtained as a bright yellow solid (167 mg, 90%) by reaction with 6h (0.150 g, 0.41 mmol).: ^1^H NMR (DMSO-d_6_, 300 MHz) δ ppm 8.00 (d, 1H, *J* = 8.6 Hz), 7.68 (d, 1H, *J* = 8.3 Hz), 7.49–7.38 (m, 5H), 7.34–7.18 (m, 8H), 6.67 (d, 1H, *J* = 15.7 Hz), 6.38–6.29 (m, 1H), 5.81 (d, NH, *J* = 3.0 Hz), 4.98–4.91 (m, 1H), 3.93 (s, 3H), 3.76–3.69 (m, 1H), 3.60 (s, 3H); ^13^C NMR (CDCl_3_, 75 MHz) δ ppm 160.90, 139.50, 136.51, 135.20, 134.54, 133.22, 130.70, 129.10, 128.96, 128.63, 128.54, 127.72, 126.47, 126.25, 124.50, 124.23, 123.94, 123.29, 119.84, 119.07, 110.08, 70.57, 63.27, 60.64, 46.36.

5,10-Dimethoxy-3-(4-methoxybenzyl)-2-phenyl-2,3-dihydrobenzo[*g*]quinazolin-4(*1H*)-one (7i). Following the procedure described above for the preparation of 7c, compound 7i was obtained as a bright yellow solid (332 mg, 98%) by reaction with 6i (0.273 g, 0.75 mmol).: ^1^H NMR (DMSO-d_6_, 300 MHz) δ ppm 8.00 (d, 1H, *J* = 8.4 Hz), 7.67 (d, 1H, *J* = 8.4 Hz), 7.46 (t, 1H, *J* = 7.7 Hz), 7.37–7.17 (m, 9H), 6.92 (d, 2H, *J* = 8.4 Hz), 5.69 (d, NH, *J* = 3.1 Hz), 5.42 (d, 1H, *J* = 15.0 Hz), 3.94 (s, 3H), 3.92 (d, 1H, *J* = 15.4 Hz), 3.73 (s, 3H), 3.57 (s, 3H); ^13^C NMR (CDCl_3_, 75 MHz) δ ppm 161.56, 159.05, 155.53, 139.47, 135.13, 134.45, 130.69, 12951, 128.97, 128.91, 128.90, 128.59, 126.08, 124.49, 123.90, 123.23, 119.79, 114.00, 110.00, 70.06, 63.24, 60.60, 55.27, 46.72.

#### 2.1.7. Synthesis of KHQ 711–713, KHQ 715–718 from 7c-7i

2,3-diphenylbenzo[*g*]quinazoline-4,5,10(*3H*)-trione (KHQ 711). To a solution of 5,10-dimethoxy-2,3-diphenyl-2,3-dihydrobenzo[g]quinazolin-4(1H)-one (50 mg, 0.12 mmol) in 1.2 mL CH_3_CN and 1.2 mL DMF, CAN (0.30 g, 0.55 mmol) in 1.2 mL distilled water was added dropwise at 0 °C for 10 min. The mixture was stirred for an additional hour at room temperature open to the air. After completion of the reaction, it was treated with ethyl acetate and water. The organic layer was separated, washed with a brine and dried over Na_2_SO_4_. After filtration, the mixture was concentrated in vacuo and purified by a silica gel flash column chromatography (50% ethyl acetate/hexane) to give compound KHQ 711 as a yellow solid (0.121 g, 45%).: mp 242.1–245.7 °C; ^1^H NMR (DMSO-d_6_, 300 MHz) δ ppm 8.15–7.84 (m, 4H), 7.46–7.28 (m, 10H); ^13^C NMR (CDCl_3_, 75 MHz) δ ppm 180.62, 158.53, 136.54, 135.11, 133.93, 133.78, 133.22, 132.00, 131.68, 130.92, 129.57, 129.30, 129.26, 128.61, 128.50, 128.19, 127.33, 127.11, 127.01, 117.39; HRMS (ESI^+^) for C_24_H_14_N_2_O_3_Na (M ^+^ •Na) calcd, 401.0897; found, 401.0899.

3-Benzyl-2-phenylbenzo[*g*]quinazoline-4,5,10(*3H*)-trione (KHQ 712). Following the procedure described above for the preparation of KHQ 711, compound KHQ 712 was obtained as a yellow solid (99 mg, 89%) by reaction with 7d (0.120 g, 0.28 mmol).: mp 214.3–216.8 °C; ^1^H NMR (DMSO-d_6_, 300 MHz) δ ppm 8.10 (d, 2H, *J* = 8.3 Hz), 7.94–7.88 (m, 2H), 7.58–7.50 (m, 5H), 7.25–7.23 (m, 3H), 7.02–6.99 (m, 2H), 5.21 (s, 2H); ^13^C NMR (CDCl_3_, 75 MHz) δ ppm 181.96, 180.81, 166.04, 158.63, 152.96, 135.10, 133.99, 133.78, 133.18, 131.64, 131.21, 128.83, 128.72, 128.69, 128.16, 128.12, 128.06, 127.52, 127.10, 126.98, 49.85; HRMS (ESI^+^) for C_25_H_16_N_2_O_3_Na (M ^+^ •Na) calcd, 415.1053; found, 415.1054.

3-Benzyl-2-p-tolylbenzo[*g*]quinazoline-4,5,10(*3H*)-trione (KHQ 716). Following the procedure described above for the preparation of KHQ 711, compound KHQ 716 was obtained as a yellow solid (93 mg, 84%) by reaction with 7e (0.120 g, 0.27 mmol).: mp 207.8–209.3 °C; ^1^H NMR (CDCl_3_, 300 MHz) δ ppm 8.29–8.22 (m, 2H), 7.84–7.77 (m, 2H), 7.38 (d, 2H, *J* = 8.2 Hz), 7.28 (d, 2H, *J* = 7.1 Hz), 7.25–7.23 (m, 3H), 7.03–7.00 (m, 2H), 5.38 (s, 2H), 2.43 (s, 3H); ^13^C NMR (CDCl_3_, 75 MHz) δ ppm 181.97, 180.77, 166.22, 158.68, 152.91, 141.80, 135.19, 135.00, 133.68, 133.15, 131.60, 131.16, 129.41, 128.66, 128.27, 127.93, 127.42, 127.01, 126.88, 116.86, 49.98, 21.49; HRMS (ESI^+^) for C_26_H_18_N_2_O_3_Na (M ^+^ •Na) calcd, 429.1210; found, 429.1213.

3-Benzyl-2-(4-fluorophenyl)benzo[*g*]quinazoline-4,5,10(*3H*)-trione (KHQ 717). Following the procedure described above for the preparation of KHQ 711, compound KHQ 717 was obtained as a yellow solid (130 mg, 75%) by reaction with 7f (0.188 g, 0.42 mmol).: mp 224.6–228.3 °C; ^1^H NMR (DMSO-d_6_, 300 MHz) δ ppm 8.10 (d, 2H, *J* = 7.5 Hz), 7.94–7.88 (m, 2H), 7.60–7.55 (m, 2H), 7.35 (t, 2H, *J* = 8.8 Hz), 7.26–7.24 (m, 3H), 7.01 (d, 2H, *J* = 7.7 Hz), 5.22 (s, 2H); ^13^C NMR (CDCl_3_, 75 MHz) δ ppm 181.81, 180.66, 165.87, 165.09, 162.52, 158.54, 152.80, 135.12, 134.99, 133.79, 133.06, 131.52, 130.66, 130.55, 130.15, 130.10, 128.81, 128.09, 127.22, 127.04, 126.92, 117.09, 116.21, 115.91, 49.91; HRMS (ESI ^+^ ) for C_25_H_15_FN_2_O_3_Na (M ^+^ •Na) calcd, 433.0959; found, 433.0960.

3-Benzyl-2-(4-methoxyphenyl)benzo[*g*]quinazoline-4,5,10(*3H*)-trione (KHQ 715). Following the procedure described above for the preparation of KHQ 711, compound KHQ715 was obtained as a yellow solid (175 mg, 79%) by reaction with 7g (0.188 g, 0.42 mmol).: mp 168.2–172.6 °C; ^1^H NMR (DMSO-d_6_, 300 MHz) δ ppm 8.09 (d, 2H, *J* = 8.1 Hz), 7.96–7.85 (m, 2H), 7.53 (d, 2H, *J* = 8.6 Hz), 7.30–7.23 (m, 3H), 7.05 (d, 4H, *J* = 8.6 Hz), 5.27 (s, 2H), 3.81 (s, 3H); ^13^C NMR (CDCl_3_, 75 MHz) δ ppm 182.12, 180.81, 165.94, 162.12, 158.92, 152.97, 135.36, 135.06, 133.70, 133.26, 131.69, 130.50, 128.79, 127.98, 127.32, 127.09, 126.94, 126.34, 116.60, 114.24, 55.57, 50.25; HRMS (ESI^+^) for C_26_H_18_N_2_O_4_Na (M ^+^ •Na) calcd, 445.1159; found, 445.1161.

3-Cinnamyl-2-phenylbenzo[*g*]quinazoline-4,5,10(*3H*)-trione (KHQ 718). Following the procedure described above for the preparation of KHQ 711, compound KHQ 718 was obtained as a yellow solid (101 mg, 79%) by reaction with 7h (0.139 g, 0.31 mmol).: mp 256.7–258.1 °C; ^1^H NMR (DMSO-d_6_, 300 MHz) δ ppm 8.11 (d, 2H, *J* = 7.7 Hz), 7.97–7.88 (m, 2H), 7.68–7.58 (m, 5H), 7.37–7.22 (m, 5H), 6.26 (d, 2H, *J* = 5.3 Hz), 4.69 (d, 2H, *J* = 3.1 Hz); ^13^C NMR (CDCl_3_, 75 MHz) δ ppm 181.94, 180.89, 165.87, 158.42, 153.00, 135.66, 135.28, 135.09, 133.93, 133.78, 133.17, 131.64, 131.26, 128.83, 128.63, 128.33, 128.14, 127.08, 127.00, 126.57, 121.52, 117.10, 49.59; HRMS (ESI ^+^ ) for C_27_H_18_N_2_O_3_Na (M ^+^ •Na) calcd, 441.1210; found, 441.1212.

3-(4-Methoxybenzyl)-2-phenylbenzo[*g*]quinazoline-4,5,10(*3H*)-trione (KHQ 713). Following the procedure described above for the preparation of KHQ 711, compound KHQ 713 was obtained as a yellow solid (0.36 g, 86%) by reaction with 7i (0.45 g, 0.98 mmol).: mp 201.7–206.3 °C; ^1^H NMR (DMSO-d_6_, 300 MHz) δ ppm 8.09 (d, 2H, *J* = 7.7 Hz), 7.94–7.87 (m, 2H), 7.60–7.52 (m, 5H), 6.90 (d, 2H, *J* = 8.6 Hz), 6.79 (d, 2H, *J* = 8.6 Hz), 5.15 (s, 2H), 3.68 (s, 3H); ^13^C NMR (CDCl_3_, 75 MHz) δ ppm 181.92, 180.80, 165.88, 159.33, 158.64, 152.86, 135.04, 134.07, 133.72, 133.13, 131.58, 131.13, 129.36, 128.82, 128.22, 127.11, 127.01, 126.91, 117.16, 113.95, 55.20, 49.23; HRMS (ESI ^+^ ) for C_26_H_18_N_2_O_4_Na (M ^+^ •Na) calcd, 445.1159; found, 445.1162.

#### 2.1.8. Synthesis of KHQ 714 from KHQ 713

2-Phenylbenzo[*g*]quinazoline-4,5,10(*3H*)-trione (KHQ 714). To a solution of 3-(4-methoxybenzyl)-2-phenylbenzo[*g*]quinazoline-4,5,10(*3H*)-trione (KHQ 713, 175 mg, 0.41 mmol) in 4 mL anhydrous DMC under argon, boron tribromide (1.2 mL, 1.2 mmol) was added at −30 °C for 10 min. After stirring at −20 °C for 1 hour, the mixture was quenched with saturated NaHCO_3_ solution at −30 °C and extracted with DMC. The organic layer was separated, washed with a brine and dried over Na_2_SO_4_. After filtration, the mixture was concentrated under reduced pressure, and purified by a silica gel flash column chromatography (66% ethyl acetate/hexane, 100% dichloromethane and then 10% methanol/dichloromethane) to give compound KHQ 714 as a yellow solid (0.079 g, 63%): mp 244 °C (decomposed); ^1^H NMR (DMSO-d_6_, 300 MHz) δ ppm 8.49 (s, 1H), 8.40–8.36 (m, 2H), 8.08–8.04 (m, 2H), 7.90–7.77 (m, 2H), 7.48–7.46 (m, 3H); HRMS (ESI^+^) for C_18_H_10_N_2_O_3_Na (M ^+^ •Na) calcd, 325.0584; found, 325.0587.

#### 2.1.9. Bacterial Strains

The *Enterococcus faecalis* (CCARM 5511), *Enterococcus faecium* (KACC 11954), *Porphyromonas gingivalis* (KCTC 5352), *Streptococcus mutans* (KACC16833), *Streptococcus sobrinus* (KCTC5809), *Staphylococcus aureus* (CCARM3506), *Staphylococcus epidermidis* (KACC 13234), *Fusobacterium nucleatum* (KCTC 2640) and *Actinomyces viscosus* (KCTC 9146) strains used in this study are listed in Table 1. *E. faecalis, E. faecium, S. epidermidis, S. aureus* and *A. viscosus* were maintained in tryptic soy broth (TSB). *S. mutans* and *S. sobrinus* were maintained in brain heart infusion broth (BHI). *P. gingivalis* was maintained in tryptic soy broth supplemented with 10% defibrinated horse blood. *F. nucleatum* was maintained in reinforced clostridial agar and incubated at 37 °C. *A. viscosus, S. sobrinus, F. nucleatum and P. gingivalis* bacteria were cultured in brain heart infusion agar plate (BHI), reinforced clostridial agar plate, tryptic soy agar plated (TSB) and blood agar plates (Blood Agar Oxoid No_2_; Oxoid, Basingstoke, UK), supplemented with 5% (*v/v*) sterile horse blood (Oxoid) at 37 °C for 24–72 h in anaerobic conditions (10% H_2_, 10% CO_2_, and balanced N_2_) [21,22,23]. 

### 2.2. Antibacterial Activity

#### 2.2.1. Broth Dilution Method

Methods for Dilution Antimicrobial Susceptibility testing developed in 2006 is the official method used in many clinical microbiology laboratories for routine antimicrobial susceptibility testing. Nowadays, many accepted and approved standards are published by the Clinical and Laboratory Standards Institute (CLSI) for bacteria and yeast testing.

Although not all fastidious bacteria can be tested accurately by this method, the standardization has been made to test certain fastidious bacterial pathogens like *Streptococci, Haemophilus influenzae, Haemophilus parainfluenzae, Neisseria gonorrhoeae* and *Neisseria meningitidis*, using specific culture media, various incubation condition [24,25]. *E. faecalis, E. faecium, S. epidermidis* and *S. aureus* were maintained in tryptic soy broth (TSB). *S. mutans* was maintained in brain heart infusion broth (BHI) and incubated at 37 °C for 24 h. *A. viscosus* was maintained in tryptic soy broth (TSB), *S. sobrinus* was maintained brain heart infusion (BHI), *F. nucleatum* was maintained in reinforced clostridial (RCM), *P. gingivalis* were cultured in tryptic soy broth (TSB) at 37 °C for 24–72 h in anaerobic conditions (10% H_2_, 10% CO_2_, and balanced N_2_) and then transferred to a 96-well plate of 0.5 McFarland standard. Serial twofold dilutions of compounds were prepared in the TSB, RCM or BHI. The MIC value was defined as the lowest compound concentration required to inhibit bacterial growth visibly after the indicated time. Oxytetracycline was used as a positive control.

The minimum bactericidal concentration was obtained using a microdilution assay according to the standard with little modification. ^5^ After bacteria were cultured in the same manner as in the MIC test, bacterial cultures were inoculated onto the agar plates and incubated at 37 °C for 24–72 h, and the bacterial colonies were counted. The lowest concentration of compound that inhibited the growth of bacteria was considered to be the minimum bactericidal concentration. The experimental and control groups were assigned to the wells in triplicate.

#### 2.2.2. Disk Diffusion Method

Broth inocula with 0.5 McFarland standard were spread onto the sterile plate by cotton swabs. P. gingivalis was cultured in tryptic soy broth supplemented with 10% defibrinated horse blood and brain heart infusion (BHI) at 37 °C for 24 h in anaerobic conditions (10% H_2_, 10% CO_2_, and balanced N_2_). Then, 8-nm sterile paper discs were impregnated with KHQ 711–718 100 μg/mL was added, and the sizes of the inhibition zones were measured after 24 h [26,27].

#### 2.2.3. Time Kill Assay

The growth curve was obtained using the previously described 96-well microplate. *P. gingivalis* was cultured in tryptic soy broth (TSB) at 37 °C for 24–72 h in anaerobic conditions (10% H_2_, 10% CO_2_, and balanced N_2_). KHQ 711–718 MIC and ½ MIC concentrations were added, and then the cells were incubated at 37 °C. Growth was evaluated by measuring OD_600_ using a microplate reader at time zero and after 6, 12, 18, 24, 30, 36, 42, 48, 54, 60, 66 and 72 h of incubation [28].

### 2.3. Statistical Analysis

All experiments were performed at least three times, and data are represented as mean ± S.D.

## 3. Results

With the pharmacophore hybridization strategy already described, a series of novel pyrimidinone or pyrimidindione-fused 1,4-naphthoquinones were synthesized (Figure 1 and Figure 2). 

A key intermediate, *N*-aryl-2-amino-1,4-dimethoxy naphthalene carboxamides 6, were synthesized from 1,4-dihydroxy-2-naphthoic acid. Esterification of 1 gave a methyl ester 2, followed by bromination of 2 by NBS to give a bromo intermediate 3. Hydrolysis of compound 3 gave an acid intermediate 4. The acid intermediate 4 was converted to various *N*-substituted amides 5 through the formation of acyl chloride intermediate and the subsequent reaction with amines. Amination of 5 with CuI/proline/sodium azide produced major intermediates 6 in good yields, but the unexpected by-products S1 were synthesized together with 6 in 22–27% yield, resulting in a decrease in the yield of 6.

Next, pyrimidin-2,4-dione-fused 1,4-naphthoquinones (KHQ 701 and KHQ 702) from the carbonylation of 6a or 6b were synthesized by triphosgene and the subsequent oxidation of 7a or 7b by ceric ammonium nitrate (CAN).

Then, pyrimidin-4-one-fused NQs were prepared from 6a-6i. Treatment of 6a-6i with various aldehydes in the presence of acid catalyst (p-TsOH∙H_2_O) gave the corresponding dihydropyrimidinone intermediates 7c-7i in high yields. The subsequent CAN oxidation of 7c-7i gave pyrimidinone NQs, KHQ 711, 712, 713, 715, 716, 717 and 718 in moderate and high yields. To prepare a N3–4-hydroxybenzyl-substituted intermediate, demethylation of KHQ 713 by BBr_3_ was attempted, but KHQ 714, a debenzylated one, was obtained instead of a 4-hydroxybenzyl-substituted one.

### 3.1. Broth Dilution Method

The minimum inhibitory concentration (MIC) values of KHQ 711, 712, 713, 714, 715, 716, 717 and 718 were 6.25–50 µg/mL against *E. faecalis* (CCARM 5511) and *E. faecium* (KACC11954), 6.25–12.5 µg/mL against *S. aureus* (CCARM 3506), 1.56–25 µg/mL against *S. epidermidis* (KACC 13234), 3.125–12.5 µg/mL against *S. mutans* (KACC16833)*,* 1.56–12.5 µg/mL against *S. sobrinus* (KCTC5809) and *P. gingivalis* (KCTC 5352), 3.125–12.5 µg/mL against *A. viscosus* (KCTC 9146) and 3.125–12.5 µg/mL against *F. nucleatum* (KCTC 2640), but the MIC values of the oxytetracycline used as a positive control were 0.195 µg/mL (KCTC 2640 and KCTC 9146), 100 µg/mL (KCTC 5352 and CCARM 5511), 1.56 µg/mL (KCTC 5809 and CCARM 3506) and 0.78 µg/mL (KCTC 5809). The MIC values of KHQ 702 were 12.5 μg/mL (KCTC 2640), 50μg/mL (KCTC 9146), 100 μg/mL (KACC16833, KCTC5809, KACC 13234 and KCTC 5352) and > 100 μg/mL (CCARM 5511, KACC11954 and CCARM 3506), respectively. [27] Oral bacteria strains exhibited resistance against KHQ 701 (MIC value > 100 μg/mL) (Table 2 and Table 3).

### 3.2. Disk Diffusion Method

The antibacterial activities of KHQ 711 (1.04 cm), 712 (1.12 cm), 713 (1.12 cm), 714 (1.2 cm), 715 (1.12 cm), 716 (1.12 cm), 717 (1.12 cm) and 718 (0.96 cm) were confirmed by measuring the diameters of the zones of growth inhibition, and the highest susceptibilities are shown in Table 4. Interestingly, the positive control, oxytetracycline (0.8 cm), showed less antibacterial activity than the fused compounds, while KHQ 714 showed the strongest activity.

### 3.3. Time Kill Assay

To confirm the antibacterial effects of the fused compounds, we constructed growth curves, and the fused compounds exhibited concentration-dependent inhibitory activities against *P. gingivalis*. Interestingly, even concentrations between 1.56–12.5 µg/mL of fused compounds completely blocked bacterial growth (Figure 3A–H), and 1.56 µg/mL of KHQ 713 showed the strongest inhibitory activity against bacteria.

## 4. Discussion

Dental disease is one of the most prevalent public health concerns. The problems caused by dental caries affect all age groups, and treatment is both expensive and labor-intensive. [1] Dental caries and periodontal diseases are infectious caused by common oral bacteria including *Lactobacillus* spp., *Streptococcus* spp. and *Actinomyces* spp., which usually form plaque biofilms on the tooth surfaces.

*E. faecalis* is an opportunistic pathogen that is frequently isolated from asymptomatic and persistent endodontic infections, especially from failed root canals undergoing retreatment [29]. *E. faecalis* is a better survivor than other root canal microbes being able to resist to many antibacterial agents [30].

Among the bacteria included in the present study, viridians *streptococci*; *S. sobrinus* and *S. mutans* were the most representative human cariogenic bacteria and are also moderately resistant to antibiotics [26]. Therefore, controlling or even reducing the levels of these pathogens is a key step in the prevention and treatment of these diseases [1,4].

Analyzing the structure-activity relationship from the results of the evaluation of the antimicrobial activities of the synthetic compounds, the pharmacophore would be a 1,4-naphthoquinone fused with a pyrimidin-4-one moiety having a phenyl group at the C2 position and a benzyl group at the N3 position.

For all bacteria strains tested, KHQ 711, 712, 713, 714, 715, 716, 717, and 718 showed the highest stable anti-bacterial activity compared with 1,4-naphthoquinone, Menadione, Juglone, Dichlon, KHQ 701 and KHQ 702 (Table 2 and Table 3). A stronger antimicrobial activity was exhibited from a 1,4-naphthoquinone derivative having a pyrimidin-4-one moiety containing both a phenyl group (C2 position) and a benzyl group (N3 position), i.e., KHQ 713 compound. This activity was more pronounced for *P. gingivalis* than for other species such as *E. faecalis* and *A. viscosus*.

Based on the antibacterial activity of synthetic derivatives, disk diffusion method and time kill assay were performed against *P. gingivalis* because *P. gingivalis* is one of the strains that exhibited strong susceptibility and is important for causing the dental caries.

The disk diffusion method has been shown that KHQ 711, 712, 713, 714, 715, 716, 717, and 718 were able to inhibit *P. gingivalis* strain growth compare to oxytetracycline (Table 3). Moreover, as shown in Figure 3A–H, all these derivatives at ½ MIC concentrations, significantly inhibited *P. gingivalis* until 18 h of incubation.

In conclusion, we synthesized 10 new pyrimidinone-fused 1,4-naphthoquinones and evaluated the antimicrobial activities of synthetic compounds against oral bacteria. Of the tested compounds, KHQ 711, 712, 713, 715, 716 and 717 showed the strongest antimicrobial activity against *P. gingivalis*. Therefore, these results suggest that these synthetic compounds with proven antimicrobial effects, KHQ 711, 712, 713, 715, 716 and 717 could be useful for the treatment of dental disease.

## Figures and Tables

**Figure 1 biomedicines-08-00160-f001:**
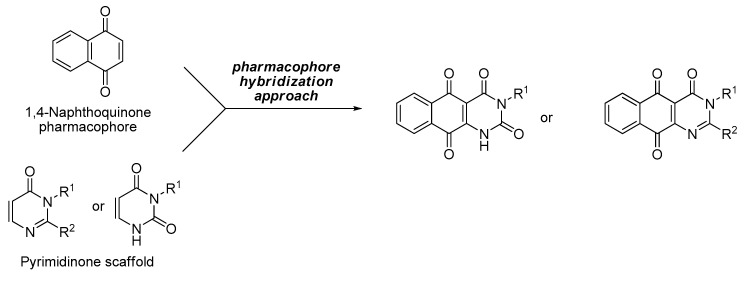
Design strategy for target compounds.

**Figure 2 biomedicines-08-00160-f002:**
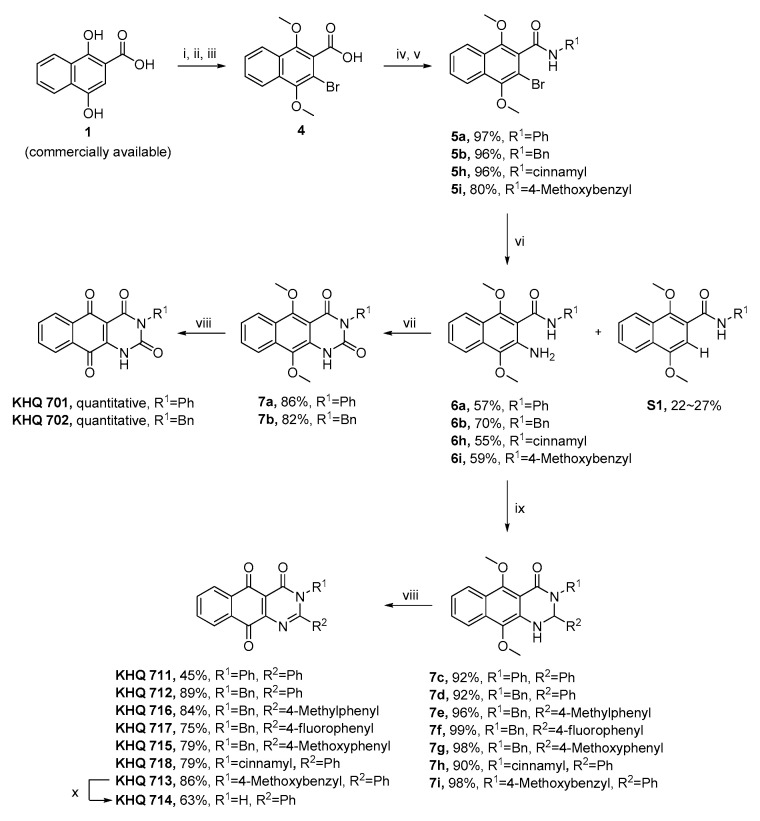
Synthesis of new pyrimidinone-fused 1,4-naphthoquinones. Reagents and conditions: (i) dimethyl sulfate, K_2_CO_3_, acetone, reflux, overnight, 93% (in the literature); (ii) NBS, DMF, rt, overnight, 98%; (iii) KOH, THF/MeOH/H_2_O (1:1:2), reflux, 2 days, 84%; (iv) oxalyl chloride, DMF, DCM, rt, 3 h; (v) R^1^-NH_2_, pyridine, DCM, rt, overnight; (vi) copper iodide, L-proline, sodium azide, DMSO, 100 °C, 3 h; (vii) triphosgene, THF, reflux, 3 h; (viii) CAN, ACN/DMF/H_2_O (1:1:1), 0 °C to rt, 1 h; (ix) R^2^-CHO, p-TsOH∙H_2_O, THF, rt, 6 h; (x) BBr_3_, DCM, −30 °C, 1 h.

**Figure 3 biomedicines-08-00160-f003:**
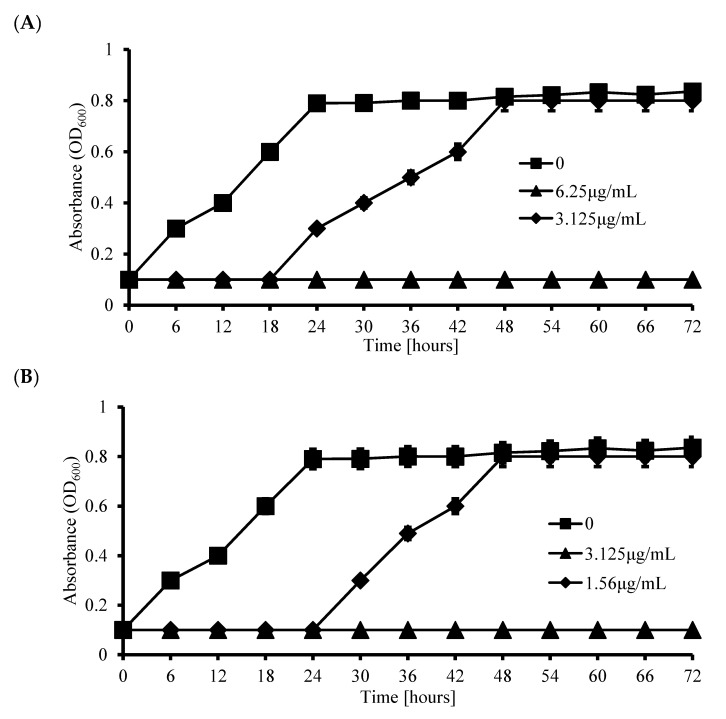
The antimicrobial activity of KHQ 711, 712, 713, 714, 715, 716, 717 and 718. *P. gingivalis* cultures were treated with (**A**) KHQ 711 (6.25 and 3.125 µg/mL), (**B**) KHQ 712 (3.125 and 1.56 µg/mL), (**C**) KHQ 713 (1.56 and 0.78 µg/mL), (**D**) KHQ 714 (12.5 and 6.25 µg/mL), (**E**) KHQ 715 (3.125 and 1.56 µg/mL), (**F**) KHQ 716 (3.125 and 1.56 µg/mL), (**G**) KHQ 717 (3.125 and 1.56 µg/mL) and (**H**) KHQ 718 (3.125 and 1.56 µg/mL) at 37 °C for 72 h and check the growth at every 6 h.

**Table 1 biomedicines-08-00160-t001:** Strains used in this study.

Strain	Description	Source
*E. faecalis*	CCARM5511	Purchased from the Korean Agricultural Culture Collection (KACC), the Culture Collection of Antimicrobial Resistant Microbes (CCARM) or the Korean Collection for Type Cultures (KCTC)
*E. faecium*	KACC11954	
*S. epidermidis*	KACC13234	
*S. mutans*	KACC16833	
*S. aureus*	CCARM3506	
*S. sobrinus*	KCTC5809	
*A. viscosus*	KCTC9146	
*F. nucleatum*	KCTC2640	
*P. gingivalis*	KCTC5352	

**Table 2 biomedicines-08-00160-t002:** Minimum inhibitory concentration (MIC) values (µg/mL) against oral bacteria.

Compound	Strains			
	*E. faecium*	*E. faecalis*	*S. aureus*	*S. epidermidis*
Oxytetracycline	0.78	100	1.56	50
1,4-naphthoquinone	12.5	100	6.25	12.5
Menadione	25	50	6.25	6.25
Juglone	>100	>100	>100	>100
Dichlon	>100	>100	25	25
KHQ 701	>100	>100	>100	>100
KHQ 702	>100	>100	>100	100
KHQ 711	6.25	6.25	6.25	6.25
KHQ 712	25	6.25	6.25	3.125
KHQ 713	6.25	6.25	6.25	1.56
KHQ 714	>100	>100	>100	25
KHQ 715	6.25	12.5	12.5	3.125
KHQ 716	12.5	12.5	12.5	1.56
KHQ 717	12.5	12.5	12.5	1.56
KHQ 718	50	25	12.5	6.25

**Table 3 biomedicines-08-00160-t003:** Minimum inhibitory concentration (MIC) values (µg/mL) against oral bacteria.

Compound	Strains				
	*S. mutans*	*A. viscosus*	*F. nucleatum*	*P. gingivalis*	*S. sobrinus*
Oxytetracycline	50	0.195	0.195	100	1.56
1,4-naphthoquinone	6.25	12.5	12.5	6.25	25
Menadione	25	25	100	12.5	50
Juglone	>100	>100	>100	>100	>100
Dichlon	>100	12.5	50	6.25	25
KHQ 701	>100	>100	>100	>100	>100
KHQ 702	100	50	12.5	100	100
KHQ 711	6.25	3.125	3.125	6.25	6.25
KHQ 712	3.125	3.125	3.125	3.125	1.56
KHQ 713	12.5	3.125	3.125	1.56	3.125
KHQ 714	6.25	12.5	6.25	12.5	12.5
KHQ 715	6.25	6.25	6.25	3.125	6.25
KHQ 716	12.5	3.125	3.125	3.125	3.125
KHQ 717	6.25	3.125	3.125	3.125	3.125
KHQ 718	12.5	6.25	6.25	3.125	12.5

**Table 4 biomedicines-08-00160-t004:** Inhibition zone diameters of KHQ 711, 712, 713, 714, 715, 716, 717 and 718 against *P. gingivalis*.

Compound	Inhibition Zone (cm) against *P. gingivalis*
Oxytetracycline	0.8
KHQ 711	1.04
KHQ 712	1.12
KHQ 713	1.12
KHQ 714	1.2
KHQ 715	1.12
KHQ 716	1.12
KHQ 717	1.12
KHQ 718	0.96

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
