# Peer review of "New Pyrimidinone-Fused 1,4-Naphthoquinone Derivatives Inhibit the Growth of Drug Resistant Oral Bacteria"

_biomedicines, 2020, doi:10.3390/biomedicines8060160_

Round 1

Reviewer 1 Report

In the current study, the authors presented the antibacterial activity of new pyrimidinine-fused 1,4-naphthoquinones against oral bacteria. The findings are of interest. However, a few points should be improved.

Comments:

The part, in the material and methods section, regarding the experimental details for synthesis of compounds should be shortened. It is too hard for the readers.

Species names should be in italics format.

Experiments presented in Table 3 should be performed also in other strains.

Line 470: the strongest activity is not observed for compound KHQ714?

Figure 3 is difficult to follow it. The quality should be improved. Maybe, also it should be splitted.

Author Response

The part, in the material and methods section, regarding the experimental details for synthesis of compounds should be shortened. It is too hard for the readers.

  • Experimental procedure for synthesizing target compounds was modified to be shorten. Also, typo errors and compound number formats were rechecked and corrected.

Species names should be in italics format.

  • Thank you for comment and we changed all of the bacterial name with italic form.

Experiments presented in Table 3 should be performed also in other strains.

  • gingivalis causes a microbial shift of the oral cavity, allowing for the uncontrolled growth of the commensal microbial community and is one of important oral bacteria that plays an important role in the onset of chronic adult periodontitis. Even though we need other bacteria for the growth test, we realized that other bacterial growth also affected by derivate with similar pattern compare with P. gingivalis based on MIC value. So we thought that the results is repetitive and not that informatic. But if reviewer want to include that results we should add one or two strain’s representative results.

Line 470: the strongest activity is not observed for compound KHQ714?

  • Even though KHQ 714 showed the antibacterial activity against mutans, A. viscosus, F. nucleatum, P. gingivalis, and S. sobrinus but did not show any activity against E. faecalis, E. faecium, and S. aureus with mild effect against S. epidermidis. So we thought that KHQ 714 possessed little or no antibacterial activity compare with other derivate.

Figure 3 is difficult to follow it. The quality should be improved. Maybe, also it should be splitted.

  • Thank you for your comment, based on that we re-made the Figure 3 with main experiment results. MIC concentration that totally block the growth of bacteria and MIC half concentration that showed dose-dependency were included and other high concentration of derivative were excluded.

Reviewer 2 Report

This work presents the synthesis and evaluation for antimicrobial activity against 9 common oral bacteria species (E. faecalis, E. faecium, S. aureus, S. epidermidis, S. mutans, S. sobrinus. P. gingivalis, A.viscosus and F. nucleatum) of 10 new pyrimidinone-fused 1,4-naphthoquinones (KHQ 701, 702, 711, 712, 713, 714, 715, 716, 717 and 718) compared to a commercial antibiotic (oxytetracycline) and four known 1,4-naphthoquinones derivates. For all strains, the susceptibility was determined by standard broth dilution method in according to CLSI while for P.gingivalis also by disk diffusion method and time kill assay.

This study revealed that 1,4-naphthoquinones having a pyrimidin-4-one moiety containing both a phenyl group in C2 position and a benzyl group at the N3 position (KHQ 711, 712, 713, 714, 715, 716, 717 and 718), were more active against all strains tested, particularly towards P. gingivalis.

The topic of the study is relevant and falling in the scopes of the journal but there are a number of general and specific problems with the manuscript ranging from readability, experimental design and flaws in the reporting of the findings. Moreover, the paper has some shortcomings in regards to some data analyses and text. I believe there is interesting data to be published from this research, but as it stands the manuscript is weak and unrefined. First of all, the title is not clear enough to identify the paper, and the abstract does not adequately summarize the conclusion drawn. Secondly, there is not balance between subsections like compounds synthesis and antimicrobial activity in material and methods. I therefore think the section needs improving in additional methodological detail about microorganisms cultivation and antimicrobial testing.

The results section needs to be improved: it is poorly organized and very superficial. Data is not well presented, it is difficult to understand what the authors mean (i.e: subsection 3.3) and, consequently, the results of the work are penalized.

From my point of view, no explanation is given about the results. Discussion should be completely reorganized and explained with more care:results need to be compared and discussed in the text with the literature data.

In addition, the English language and the clarity of the whole paper require more revision. I would suggest letting a native English speaker to carry out language edition for the manuscript.

In my opinion, the manuscript is not acceptable for publication since several modifications are necessary before it can be considered for publications.

Some specific comments

Title

The title could be completed by mentioning the synthesis of compounds.

Abstract

- lines 12-15: please add names of bacterial species and abbreviations of all 1,4-naphthoquinones
tested

- lines 27-30: conclusions are too vague and generic

Introduction

page 2

- lines 62-65: please move this sentence in the discussion section

Material and methods

pages 8-9

- lines 388-395: please mark all microbial names in Italic font

- line393: please add “broth” after “brain heart infusion”

- line 394: please delete “agar” after “horse blood”

- lines 392-395:please add bacterial growth conditions of different species (i.e. P.gingivalis and
F.nucleatum are obligate anaerobic bacteria while Streptococcus spp. are facultative anaerobic
bacteria)

- line 397: please change subtitle “2.1.10” to “2.2” and delete “of KHQ against oral bacteria”

- line 398: please add subtitle “2.2.1 Broth dilution method”

- line 400: please change “Muller” to “Müeller”

- line 406: please explain what modification of standard method were introduced by authors

- line 411: please change subtitle “2.2” to “2.2.2” and change “Susceptibility test” to“Disk diffusion
method”

- line 412: please change “inoculums” to “inocula” (it's a word of Latin origin, not English!)

- line 416: please change subtitle “2.3” to “2.2.3” and change “Bacterial growth test” to“Time kill
assay”

- lines 411-416 (subsections 2.2 and 2.3): please add bacterial growth conditions of different species (temperature, aerobe/anaerobe conditions, time points of incubation...) and explain what the concentrations of 1,4-naphthoquinones were used in this assays

Table 1

- please exchange KCTC 2640 with KCTC 9146 in the column of description

Results

pages 11-12

- lines 449, 464 and 471: please change subtitles of subsections 3.1, 3.2 and 3.3 to “Broth dilution method”, “Disk diffusion method” and “Time kill assay”,respectively. Usually, no results information were reported in subtitles.

- lines 450-451: please delete sentence “Antibacterial susceptibility....microdilution method”.

It is redundant.

- lines 454-456: please change values of 3.125 -100 µg/ml to 3.125-12.15µg/ml,1.56- 100 µg/ml to 1.56-12.15 µg/ml, 3.125 -50 µg/ml to 3.125-12.15µg/ml and 3.125-12.15µg/ml to 3.125-6.25µg/ml, respectively.

- line 458: please change KACC 11954 to KCTC 5809

- line 459: please change “Tables 2 and 3” to “Table 2A and 2B”.

Subsections 3.1, 3.2 and 3.3

- Why did not authors speak about the results of KHQ 701 and 702? Why author did not make a comparison among new and known 1,4-naphthoquinones derivates?

- What was the reason for the choice of menadione, juglone and dichlon? Menadione is a coagulant, juglone is occasionally used as a herbicide and dichlon as a fungicide. No detailed information is given.

- Why did authors make disk diffusion method and time kill assay only for P.aeruginosa? New 1,4-
naphthoquinones derivates also seem to act well towards
S.sobrinus (Table 2)...

- lines 465-466: please delete sentence “A bacterial susceptibility....solid agar plates”.It is
redundant.

- lines 474-476: please explain these sentences. Why did 1,56 µg/ml of KHQ713 show the strongest
inhibitory activity? Figure 3 cannot be easily understood
:curves are overlapping and it is not clear
which concentrations the curves refer to.

page 14

The figure caption is very long and unclear: some information are redundant.

Discussion

page 17

Discussion should be reorganized and explained with more care, because it contains a lot of repetitive information. The data contained could enlarged our knowledge but it is not well explicated the medical impact and how will mean to address future experiments.

Author Response

Title

The title could be completed by mentioning the synthesis of compounds.

  • New Pyrimidinone-Fused 1,4-Naphthoquinones Derivates Inhibit the Growth of Drug Resistant Oral Bacteria.

 Abstract

lines 12-15: please add names of bacterial species and abbreviations of all 1,4-naphthoquinones tested

  • Thank you for comment and we changed

Methods/Principal Findings: In this study, new 10 pyrimidinone-fused 1,4-naphthoquinones including KHQ 711, 712, 713, 714, 715, 716, 717 and 718 were synthesized from simple starting compounds and four known 1,4-naphthoquinones were evaluated for antimicrobial activity against Enterococcus faecalis, Enterococcus faecium, Staphylococcus aureus, Staphylococcus epidermidis, Streptococcus mutans, Streptococcus sobrinus, Porphyromonas gingivalis, Actinomyces viscosus and Fusobacterium nucleatum. Pyrimidinone-fused 1,4-naphthoquinones were synthesized in good yields through a series of chemical reactions from a commercially available 1,4-dihydroxynaphthoic acid. MIC values of KHQ 711, 712, 713, 714, 715, 716, 717 and 718 were 6.25-50 μg/mL against E. faecalis (CCARM 5511), 6.25-25 μg/mL against E. faecium (KCTC 5809) and S. aureus (CCARM 3506), 1.56-25 μg/mL against S. epidermidis (KACC 13234), 3.125-100 μg/mL against S. mutans (KACC16833), 1.56-100 μg/mL against S. sobrinus (KCTC5809) and P. gingivalis (KCTC 5352), 3.125-50 μg/mL against A. viscosus (KCTC 9146) and 3.125-12.5 μg/mL against F. nucleatum (KCTC 2640) with a broth microdilution assay. A disk diffusion assay with KHQ derivatives were also exhibited strong susceptibility with inhibition zones with 0.96 to 1.2 cm in size against P. gingivalis. Among the 10 compounds evaluated, KHQ 711, 712, 713, 715, 716 and 717 demonstrated strong antimicrobial activity against the 9 types of pathogenic oral bacteria. A pyrimidin-4-one moiety comprising a phenyl group at the C-2 position and a benzyl group at the N3 position appears to be essential for physiological activity.

 lines 27-30: conclusions are too vague and generic

  • Thank you for comment and we changed

Conclusion/Significance: Pyrimidinone-fused 1,4-naphthoquinones synthesized from simple starting compounds and four known 1,4-naphthoquinones were synthesized and showed strong antibacterial activity to the 9 common oral bacteria. These results suggest that these derivate should be prospective for the treatment of dental diseases caused by oral bacteria including drug-resistant strains.

Introduction

lines 62-65: please move this sentence in the discussion section

  • Thank you for comment and we changed

Two major dental diseases in the world are dental caries and periodontal disease, both of which are caused by various bacteria in the oral cavity. 19 Dental caries is a common oral disease that usually develops the formation of plaque biofilms on the tooth surfaces, and the causative agents are Gram-positive bacteria such as Streptococcus mutans and Streptococcus sobrinus as well as some non-mutans streptococci.17, 27 Specific bacterial species such as Actinomyces spp. and Enterococcus faecalis contribute to tooth root caries and periodontal infection. 18, 27

Although dental disease is only slowly progressive, oral bacteria can also cause infections of the head and neck in locations such as periapical abscesses, the jaw bones and fascia. 2 Therefore, the control of oral bacteria is a key to the prevention and the treatment of these oral diseases. Various antibiotics including ampicillin, chlorhexidine, erythromycin, spiramycin and vancomycin have been used to prevent dental caries, but these agents can cause unexpected side effects such as microorganism resistance, vomiting and diarrhea. 2, 8 Furthermore, the use of antibiotics can promote the development of multidrug-resistant (MDR) strains of bacteria. 7 These problems have led to a search for new antibacterial substances that are specific to oral pathogens. 4

1,4-Naphthoquinone (NQ) is the central chemical structure of natural compounds such as menadione and juglone, and it plays crucial roles in the development and innovation of new drugs due to its various pharmacological properties such as antibacterial, antifungal, antiviral, insecticidal and anti-inflammatory activity. Among the 1,4-naphthoquinone derivatives, heterocycle-fused naphthoquinones in particular have been demonstrated to exhibit a variety of biological activities. 1, 12, 21-23

Pyrimidine derivatives, such as pyrimidin-4-one and pyrimidine-2,4-dione, have also various biological activities such as antimicrobial, antiviral and antifungal activities, and thus are considered as a major component for drug discovery. 14, 16

The conjugation of pyrimidinone, which exhibits antimicrobial activity, to 1,4-naphthoquinone, which exhibits antimicrobial, antifungal and antiviral activity, could lead to the development of synthetic antimicrobial drugs having better activity.

With the above considerations, a series of novel pyrimidinone or pyrimidindione-fused 1,4-naphthoquinones were synthesized via a pharmacophore hybridization strategy. 10

Material and methods

lines 388-395: please mark all microbial names in Italic font

  • Thank you for comment and we changed

The Enterococcus faecalis (CCARM 5511), Enterococcus faecium (KCTC 5809), Porphyromonas gingivalis (KCTC 5352), Streptococcus mutans (KACC16833), Streptococcus sobrinus (KCTC5809), Staphylococcus aureus (CCARM3506), Staphylococcus epidermidis (KACC 13234), Fusobacterium nucleatum (KCTC 2640) and Actinomyces viscosus (KCTC 9146) strains used in this study are listed in Table 1. E. faecalis, E. faecium, S. epidermidis, S. aureus and A. viscosus were maintained in tryptic soy broth (TSB). S. mutans and S. sobrinus were maintained in brain heart infusion (BHI). P. gingivalis was maintained in tryptic soy broth supplemented with 10% defibrinated horse blood. F. nucleatum was maintained in reinforced clostridial agar and incubated at 37. A. viscosus, S. sobrinus, F. nucleatum and P. gingivalis bacteria were cultured in brain heart infusion agar plate (BHI), reinforced clostridial agar plate, tryptic soy agar plated (TSB) and blood agar plates (Blood Agar Oxoid No2; Oxoid, Basingstoke, UK), supplemented with 5% (v/v) sterile horse blood (Oxoid) at 37 °C for 24–72 h in anaerobic conditions (10% H2, 10% CO2, and balanced N2). 11, 15, 29

line 393: please add “broth” after “brain heart infusion”

  • Thank you for comment and we changed

The Enterococcus faecalis (CCARM 5511), Enterococcus faecium (KCTC 5809), Porphyromonas gingivalis (KCTC 5352), Streptococcus mutans (KACC16833), Streptococcus sobrinus (KCTC5809), Staphylococcus aureus (CCARM3506), Staphylococcus epidermidis (KACC 13234), Fusobacterium nucleatum (KCTC 2640) and Actinomyces viscosus (KCTC 9146) strains used in this study are listed in Table 1. E. faecalis, E. faecium, S. epidermidis, S. aureus and A. viscosus were maintained in tryptic soy broth (TSB). S. mutans and S. sobrinus were maintained in brain heart infusion broth (BHI). P. gingivalis was maintained in tryptic soy broth supplemented with 10% defibrinated horse blood. F. nucleatum was maintained in reinforced clostridial agar and incubated at 37. A. viscosus, S. sobrinus, F. nucleatum and P. gingivalis bacteria were cultured in brain heart infusion agar plate (BHI), reinforced clostridial agar plate, tryptic soy agar plated (TSB) and blood agar plates (Blood Agar Oxoid No2; Oxoid, Basingstoke, UK), supplemented with 5% (v/v) sterile horse blood (Oxoid) at 37 °C for 24–72 h in anaerobic conditions (10% H2, 10% CO2, and balanced N2). 11, 15, 29

line 394: please delete “agar” after “horse blood”

  • Thank you for comment and we changed

The Enterococcus faecalis (CCARM 5511), Enterococcus faecium (KCTC 5809), Porphyromonas gingivalis (KCTC 5352), Streptococcus mutans (KACC16833), Streptococcus sobrinus (KCTC5809), Staphylococcus aureus (CCARM3506), Staphylococcus epidermidis (KACC 13234), Fusobacterium nucleatum (KCTC 2640) and Actinomyces viscosus (KCTC 9146) strains used in this study are listed in Table 1. E. faecalis, E. faecium, S. epidermidis, S. aureus and A. viscosus were maintained in tryptic soy broth (TSB). S. mutans and S. sobrinus were maintained in brain heart infusion broth (BHI). P. gingivalis was maintained in tryptic soy broth supplemented with 10% defibrinated horse blood. F. nucleatum was maintained in reinforced clostridial agar and incubated at 37. A. viscosus, S. sobrinus, F. nucleatum and P. gingivalis bacteria were cultured in brain heart infusion agar plate (BHI), reinforced clostridial agar plate, tryptic soy agar plated (TSB) and blood agar plates (Blood Agar Oxoid No2; Oxoid, Basingstoke, UK), supplemented with 5% (v/v) sterile horse blood (Oxoid) at 37 °C for 24–72 h in anaerobic conditions (10% H2, 10% CO2, and balanced N2). 11, 15, 29

lines 392-395: please add bacterial growth conditions of different species (i.e. P.gingivalis and F.nucleatum are obligate anaerobic bacteria while Streptococcus spp. are facultative anaerobic bacteria)

  • Thank you for comment and we changed

The Enterococcus faecalis (CCARM 5511), Enterococcus faecium (KCTC 5809), Porphyromonas gingivalis (KCTC 5352), Streptococcus mutans (KACC16833), Streptococcus sobrinus (KCTC5809), Staphylococcus aureus (CCARM3506), Staphylococcus epidermidis (KACC 13234), Fusobacterium nucleatum (KCTC 2640) and Actinomyces viscosus (KCTC 9146) strains used in this study are listed in Table 1. E. faecalis, E. faecium, S. epidermidis, S. aureus and A. viscosus were maintained in tryptic soy broth (TSB). S. mutans and S. sobrinus were maintained in brain heart infusion broth (BHI). P. gingivalis was maintained in tryptic soy broth supplemented with 10% defibrinated horse blood. F. nucleatum was maintained in reinforced clostridial agar and incubated at 37. A. viscosus, S. sobrinus, F. nucleatum and P. gingivalis bacteria were cultured in brain heart infusion agar plate (BHI), reinforced clostridial agar plate, tryptic soy agar plated (TSB) and blood agar plates (Blood Agar Oxoid No2; Oxoid, Basingstoke, UK), supplemented with 5% (v/v) sterile horse blood (Oxoid) at 37 °C for 24–72 h in anaerobic conditions (10% H2, 10% CO2, and balanced N2). 11, 15, 29

line 397: please change subtitle “2.1.10” to “2.2” and delete “of KHQ against oral bacteria”

  • Thank you for comment and we changed

2.2. Antibacterial activity

2.2.1 Broth dilution method

The minimum inhibitory concentrations (MICs) were determined based on the broth microdilution method described in the Clinical and Laboratory Standards Institute 2006 guidelines using microplates. 5,24 E. faecalis, E. faecium, S. epidermidis, S. aureus were maintained in tryptic soy broth (TSB). S. mutans was maintained in brain heart infusion broth (BHI) and incubated at 37℃ for 24hours.

  1. viscosus was maintained tryptic soy agar broth (TSB), S. sobrinus was maintained brain heart infusion agar plate (BHI), F. nucleatum was maintained in reinforced clostridial (RCM), P. gingivalis were cultured in tryptic soy broth at 37 °C for 24–72 h in anaerobic conditions (10% H2, 10% CO2, and balanced N2) and then transferred to a 96-well plate of 0.5 McFarland standard. Serial twofold dilutions of compounds were prepared in the TSB, RCM the BHI. The MIC value was defined as the lowest compound concentration required to inhibit bacterial growth visibly after the indicated time. Oxytetracycline was used as a positive control.

The minimum bactericidal concentration was obtained using a microdilution assay according to the standard with little modification. 5 After bacteria were cultured in the same manner as in the MIC test, bacterial cultures were inoculated onto the agar plates and incubated at 37 °C for 24-72 hours, and the bacterial colonies were counted. The lowest concentration of compound that inhibited the growth of bacteria was considered as the minimum bactericidal concentration. The experimental and control groups were assigned to the wells in triplicate.

line 398: please add subtitle “2.2.1 Broth dilution method”

  • Thank you for comment and we changed

2.2. Antibacterial activity

2.2.1 Broth dilution method

The minimum inhibitory concentrations (MICs) were determined based on the broth microdilution method described in the Clinical and Laboratory Standards Institute 2006 guidelines using microplates. 5,24 E. faecalis, E. faecium, S. epidermidis, S. aureus were maintained in tryptic soy broth (TSB). S. mutans was maintained in brain heart infusion broth (BHI) and incubated at 37℃ for 24hours.

  1. viscosus was maintained tryptic soy broth (TSB), S. sobrinus was maintained brain heart infusion (BHI), F. nucleatum was maintained in reinforced clostridial (RCM), P. gingivalis were cultured in tryptic soy broth (TSB) at 37 °C for 24–72 h in anaerobic conditions (10% H2, 10% CO2, and balanced N2) and then transferred to a 96-well plate of 0.5 McFarland standard. Serial twofold dilutions of compounds were prepared in the TSB, RCM the BHI.

The MIC value was defined as the lowest compound concentration required to inhibit bacterial growth visibly after the indicated time. Oxytetracycline was used as a positive control.

The minimum bactericidal concentration was obtained using a microdilution assay according to the standard with little modification. 5 After bacteria were cultured in the same manner as in the MIC test, bacterial cultures were inoculated onto the agar plates and incubated at 37 °C for 24-72 hours, and the bacterial colonies were counted. The lowest concentration of compound that inhibited the growth of bacteria was considered as the minimum bactericidal concentration. The experimental and control groups were assigned to the wells in triplicate.

line 400: please change “Muller” to “Müeller”

  • Thank you for comment and we changed

line 406: please explain what modification of standard method were introduced by authors.

  • Thank you for comment and we changed

Methods for Dilution Antimicrobial Susceptibility testing developed in 2006 is the official method used in many clinical microbiology laboratories for routine antimicrobial susceptibility testing. Nowadays, many accepted and approved standards are published by the Clinical and Laboratory Standards Institute (CLSI) for bacteria and yeast testing.

Although not all fastidious bacteria can be tested accurately by this method, the standardization has been made to test certain fastidious bacterial pathogens like streptococci, Haemophilus influenzae, Haemophilus parainfluenzae, Neisseria gonorrhoeae and Neisseria meningitidis, using specific culture media, various incubation conditions.

line 411: please change subtitle “2.2” to “2.2.2” and change “Susceptibility test” to“Disk diffusion method”

  • Thank you for comment and we changed

2.2.2 Disk diffusion method

Broth inocula with 0.5 McFarland standard were spread onto the sterile P. gingivalis were cultured in tryptic soy broth supplemented with 10% defibrinated horse blood at 37 °C for 24–72 h in anaerobic conditions (10% H2, 10% CO2, and balanced N2) by cotton swabs. Then, 8-nm sterile paper discs were impregnated with KHQ 711-718 100μg/mL was added, and the sizes of the inhibition zones were measured after 24-72 hours. 6, 9

line 412: please change “inoculums” to “inocula” (it's a word of Latin origin, not English!)

  • Thank you for comment and we changed

Broth inocula with 0.5 McFarland standard were spread onto the sterile P. gingivalis were cultured in tryptic soy broth supplemented with 10% defibrinated horse blood at 37 °C for 24–72 h in anaerobic conditions (10% H2, 10% CO2, and balanced N2) by cotton swabs. Then, 8-nm sterile paper discs were impregnated with KHQ 711-718 100μg/mL was added, and the sizes of the inhibition zones were measured after 24-72 hours. 6, 9

 line 416: please change subtitle “2.3” to “2.2.3” and change “Bacterial growth test” to“Time kill assay”

2.2.3. Time kill assay.

  • Thank you for comment and we changed

The growth curve was obtained using the previously described 96-well microplate. P. gingivalis were cultured in tryptic soy broth (TSB) 37 °C for 24–72 h in anaerobic conditions (10% H2, 10% CO2, and balanced N2). KHQ 711-718(1.56-12.5μg/mL) was added, and then the cells were incubated at 37℃. Growth was evaluated by measuring OD600 using a microplate reader after 0, 6, 12, 18, 24, 30, 36, 42, 48, 54, 60, 66 and 72 h.13

lines 411-416 (subsections 2.2 and 2.3): please add bacterial growth conditions of different species (temperature, aerobe/anaerobe conditions, time points of incubation...) and explain what the concentrations of 1,4-naphthoquinones were used in this assays.

  • Thank you for comment and we changed

2.2.2 Disk diffusion method

Broth inocula with 0.5 McFarland standard were spread onto the sterile P. gingivalis were cultured in tryptic soy broth supplemented with 10% defibrinated horse blood at 37 °C for 24–72 h in anaerobic conditions (10% H2, 10% CO2, and balanced N2) by cotton swabs. Then, 8-nm sterile paper discs were impregnated with KHQ 711-718 100μg/mL was added, and the sizes of the inhibition zones were measured after 24-72 hours. 6, 9

2.2.3. Time kill assay.

The growth curve was obtained using the previously described 96-well microplate. P. gingivalis were cultured in tryptic soy broth (TSB) 37 °C for 24–72 h in anaerobic conditions (10% H2, 10% CO2, and balanced N2). KHQ 711-718(1.56-12.5μg/mL) was added, and then the cells were incubated at 37℃. Growth was evaluated by measuring OD600 using a microplate reader after 0, 6, 12, 18, 24, 30, 36, 42, 48, 54, 60, 66 and 72 h.13

 Table 1

please exchange KCTC 2640 with KCTC 9146 in the column of description

  • Thank you for comment and we changed

Results

lines 449, 464 and 471: please change subtitles of subsections 3.1, 3.2 and 3.3 to “Broth dilution method”, “Disk diffusion method” and “Time kill assay”,respectively. Usually, no results information were reported in subtitles.

  • Thank you for comment and we changed

3.1. Broth dilution method

The minimum inhibitory concentration (MIC) values of KHQ 711, 712, 713, 714, 715, 716, 717 and 718 were 6.25-50 µg/mL against E. faecalis (CCARM 5511) and E. faecium (KCTC 5809), 6.25-12.5 µg/mL against S. aureus (CCARM 3506), 1.56-25 µg/mL against S. epidermidis (KACC 13234), 3.125-12.5 µg/mL against S. mutans (KACC16833), 1.56-12.5 µg/mL against S. sobrinus (KCTC5809) and P. gingivalis (KCTC 5352), 3.125-12.5µg/mL against A. viscosus (KCTC 9146) and 3.125-12.5 µg/mL against F. nucleatum (KCTC 2640), but the MIC values of the oxytetracycline used as a positive control were 0.195 µg/mL (KCTC 2640 and KCTC 9146), 100 µg/mL (KCTC 5352 and CCARM 5511), 1.56 µg/mL (KCTC 5809 and CCARM 3506) and 0.78 μg/mL (KCTC 5809) (Tables 2A and 2B).

3.2. Disk diffusion method

The antibacterial activities of KHQ 711 (1.04 cm), 712 (1.12 cm), 713 (1.2 cm), 714 (1.12 cm), 715 (1.12 cm), 716 (1.12 cm), 717 (1.12 cm) and 718 (0.98 cm) were confirmed by measuring the diameters of the zones of growth inhibition, and the highest susceptibilities are shown in Table 3. Interestingly, the positive control, oxytetracycline (0.8 cm), showed less antibacterial activity than the fused compounds, while KHQ713 showed the strongest activity.

3.3. Time kill assay

To confirm the antibacterial effects of the fused compounds, we constructed growth curves, and the fused compounds exhibited concentration-dependent inhibitory activities against P. gingivalis. Interestingly, even concentrations between 1.56-12.5 µg/mL of fused compounds completely blocked bacterial growth (Figure 3A-H), and 1.56 µg/mL of KHQ713 showed the strongest inhibitory activity against bacteria.

lines 450-451: please delete sentence “Antibacterial susceptibility....microdilution method”. It is redundant.

  • Thank you for comment and we changed

3.1. Broth dilution method

The minimum inhibitory concentration (MIC) values of KHQ 711, 712, 713, 714, 715, 716, 717 and 718 were 6.25-50 µg/mL against E. faecalis (CCARM 5511) and E. faecium (KCTC 5809), 6.25-12.5 µg/mL against S. aureus (CCARM 3506), 1.56-25 µg/mL against S. epidermidis (KACC 13234), 3.125-12.5 µg/mL against S. mutans (KACC16833), 1.56-12.5 µg/mL against S. sobrinus (KCTC5809) and P. gingivalis (KCTC 5352), 3.125-12.5µg/mL against A. viscosus (KCTC 9146) and 3.125-12.5 µg/mL against F. nucleatum (KCTC 2640), but the MIC values of the oxytetracycline used as a positive control were 0.195 µg/mL (KCTC 2640 and KCTC 9146), 100 µg/mL (KCTC 5352 and CCARM 5511), 1.56 µg/mL (KCTC 5809 and CCARM 3506) and 0.78 μg/mL (KCTC 5809) (Table 2A and 2B).

lines 454-456: please change values of 3.125 -100 µg/ml to 3.125-12.15µg/ml,1.56- 100 µg/ml to 1.56-12.15 µg/ml, 3.125 -50 µg/ml to 3.125-12.15µg/ml and 3.125-12.15µg/ml to 3.125-6.25µg/ml, respectively.

  • Thank you for comment and we changed

3.1. Broth dilution method

The minimum inhibitory concentration (MIC) values of KHQ 711, 712, 713, 714, 715, 716, 717 and 718 were 6.25-50 µg/mL against E. faecalis (CCARM 5511) and E. faecium (KCTC 5809), 6.25-12.5 µg/mL against S. aureus (CCARM 3506), 1.56-25 µg/mL against S. epidermidis (KACC 13234), 3.125-12.5 µg/mL against S. mutans (KACC16833), 1.56-12.5 µg/mL against S. sobrinus (KCTC5809) and P. gingivalis (KCTC 5352), 3.125-12.5µg/mL against A. viscosus (KCTC 9146) and 3.125-12.5 µg/mL against F. nucleatum (KCTC 2640), but the MIC values of the oxytetracycline used as a positive control were 0.195 µg/mL (KCTC 2640 and KCTC 9146), 100 µg/mL (KCTC 5352 and CCARM 5511), 1.56 µg/mL (KCTC 5809 and CCARM 3506) and 0.78 μg/mL (KCTC 5809) (Table 2A and 2B).

line 458: please change KACC 11954 to KCTC 5809

  • Thank you for comment and we changed

 line 459: please change “Tables 2 and 3” to “Table 2A and 2B”.

  • Thank you for comment and we changed

The minimum inhibitory concentration (MIC) values of KHQ 711, 712, 713, 714, 715, 716, 717 and 718 were 6.25-50 µg/mL against E. faecalis (CCARM 5511) and E. faecium (KCTC 5809), 6.25-12.5 µg/mL against S. aureus (CCARM 3506), 1.56-25 µg/mL against S. epidermidis (KACC 13234), 3.125-12.5 µg/mL against S. mutans (KACC16833), 1.56-12.5 µg/mL against S. sobrinus (KCTC5809) and P. gingivalis (KCTC 5352), 3.125-12.5µg/mL against A. viscosus (KCTC 9146) and 3.125-12.5 µg/mL against F. nucleatum (KCTC 2640), but the MIC values of the oxytetracycline used as a positive control were 0.195 µg/mL (KCTC 2640 and KCTC 9146), 100 µg/mL (KCTC 5352 and CCARM 5511), 1.56 µg/mL (KCTC 5809 and CCARM 3506) and 0.78 μg/mL (KCTC 5809) (Table 2A and 2B).

Subsections 3.1, 3.2 and 3.3

Why did not authors speak about the results of KHQ 701 and 702? Why author did not make a comparison among new and known 1,4-naphthoquinones derivates?

  • Thank you for comment and we changed

The MIC values of KHQ 702 used 12.5/mL (KCTC 2640), 50/mL (KCTC 9146), 100/mL (KACC16833, KCTC5809 and KCTC 5352) and >100/mL (CCARM 5511, KCTC 5809, CCARM 3506 and KACC 13234), respectively. 9 oral bacteria strains exhibited resistance against KHQ 701 (MIC value >100/mL) (Table 2A,B).

What was the reason for the choice of menadione, juglone and dichlon? Menadione is a coagulant, juglone is occasionally used as a herbicide and dichlon as a fungicide. No detailed information is given.

  • Thank you for your comment. In this introduction part, I would like to introduce and emphasize 1,4-naphtnoquinone moiety is biologically important. I just commented dichlon also and modified this parts as shown below,

“1,4-Naphthoquinone (1,4-NQ) is the central chemical structure of biologically active compounds such as menadione as a coagulant, juglone as a herbicide, and dichlone as a fungicide. As such, the 1,4-NQ moiety is known as an important pharmacophore that exhibits various pharmacological properties such as antibacterial, antifungal, antiviral, pesticide and anti-inflammatory activities, and thus plays an important role in the development of new drugs. Among the 1,4-naphthoquinone derivatives, heterocycle-fused naphthoquinones in particular have been demonstrated to exhibit a variety of biological activities. 1, 12, 21-23

Why did authors make disk diffusion method and time kill assay only for P.gingivalis?

  • Thank you for comment

New 1,4-naphthoquinones derivates also seem to act well towards S. sobrinus (Table 2) but we thought P. gingivalis is more important than S. sobrinus, because P. gingivalis secretes toxic products that contribute to the inactivation of the effector molecules of the host immune response leading to tissue destruction, and if we include all of the time kill assay results it should be too much with redundant results. SO we only include present the P. gingivalis time kill assay results as a representative.

lines 465-466: please delete sentence “A bacterial susceptibility....solid agar plates”. It is redundant.

  • Thank you for comment and we changed

The minimum inhibitory concentration (MIC) values of KHQ 711, 712, 713, 714, 715, 716, 717 and 718 were 6.25-50 µg/mL against E. faecalis (CCARM 5511) and E. faecium (KCTC 5809), 6.25-12.5 µg/mL against S. aureus (CCARM 3506), 1.56-25 µg/mL against S. epidermidis (KACC 13234), 3.125-12.5 µg/mL against S. mutans (KACC16833), 1.56-12.5 µg/mL against S. sobrinus (KCTC5809) and P. gingivalis (KCTC 5352), 3.125-12.5µg/mL against A. viscosus (KCTC 9146) and 3.125-12.5 µg/mL against F. nucleatum (KCTC 2640), but the MIC values of the oxytetracycline used as a positive control were 0.195 µg/mL (KCTC 2640 and KCTC 9146), 100 µg/mL (KCTC 5352 and CCARM 5511), 1.56 µg/mL (KCTC 5809 and CCARM 3506) and 0.78 μg/mL (KCTC 5809) (Table 2A and 2B).

lines 474-476: please explain these sentences. Why did 1,56 µg/ml of KHQ713 show the strongest inhibitory activity? Figure 3 cannot be easily understood:curves are overlapping and it is not clear which concentrations the curves refer to.

  • Thank you for comment

KHQ713 showed the strongest antibacterial activity compare with tested derivate. So changed “Interestingly, even concentrations between 0.78-12.5 µg/mL of fused compounds completely blocked bacterial growth (Figure 3A-H), and KHQ713 showed the strongest bacterial growth inhibitory activity against P. gingivalis.”

We found manuscript formatting is changed during change the original manuscript to the publishable format. So we changed the pictures simply to be seen more clearly.

Page 14

The figure caption is very long and unclear: some information are redundant.

Figure 1 Designing strategy of target compounds in this study

Figure 2 Chemical structure of 1,4 naphthoquinone derivatives

  • Thank you for comment. We changed Figure 1 and its position, and removed Figure 2.

 Figure 2 The antimicrobial activity of KHQ 711, 712, 713, 714, 715, 716, 727 and 718 inhibited growth of P. gingivalis. P. gingivalis cultures were treated with (A) KHQ 711(6.25 and 3.125 µg/mL), (B) KHQ 712 (3.125 and 1.56µg/mL), (C) KHQ 713(1.56 and 0.78µg/mL), (D) KHQ 714(12.5 and 6.25µg/mL), (E) KHQ 715(3.125 and 1.56µg/mL), (F) KHQ 716(3.125 and 1.56µg/mL), (G) KHQ 717(3.125 and 1.56µg/mL) and (H) KHQ 718 (3.125 and 1.56µg/mL) at 37 for 72 hours and check the growth at every 6 hours.

 (A)(B)(C)(D)(E)(F)(G)(H)

 Discussion

Discussion should be reorganized and explained with more care, because it contains a lot of repetitive information. The data contained could enlarged our knowledge but it is not well explicated the medical impact and how will mean to address future experiments.

  • Thank you for comment and we changed

Oral bacteria such as E. faecalis, E. faecium, S. mutans, S. sobrinus, S. aureus, S. epidermidis, P. gingivalis, A. viscosus, and F. nucleatum contribute to dental caries and periodontal diseases including the formation of plaque biofilms on the tooth surfaces. 18,20,25-27, and show a high level of resistance to commonly used antibiotics like tetracycline, methicillin, oxacillin, and vancomycin. 27 The development of the compound to reduce the issue of drug resistance is urgent.

Analyzing the structure and activity relationship from the results of the evaluation of the antimicrobial activities of the synthetic compounds, the pharmacophore for showing activity would be a 1,4-naphthoquinone having a pyrimidin-4-one moiety containing both a phenyl group (C2 position) and a benzyl group (N3 position).

KHQ 711, 712, 713, 714, 715, 716, 717, and 718 showed the highest stable anti-bacterial activity compared with 1,4-naphthoquinone, Menadione, Juglone, Dichlon, KHQ 701, and KHQ 702 which were used to treat E. faecalis, E. faecium, S. mutans, S. sobrinus, S. aureus, S. epidermidis, P. gingivalis, A. viscosus, and F. nucleatum (Table 2A and 2B).

The disk diffusion method also showed KHQ 711, 712, 713, 714, 715, 716, 717, and 718 strongly inhibited the oral bacterial growth compared with oxytetracycline (Table 3).

In conclusion, we synthesized 10 new pyrimidinone-fused 1,4-naphthoquinones and evaluated the antimicrobial activities of synthetic compounds against oral bacteria. Of the tested compounds, KHQ 711, 712, 713, 715, 716 and 717 showed the strongest antimicrobial activity against P. gingivalis. Therefore, these results suggest that these synthetic compounds with proven antimicrobial effects, KHQ 711, 712, 713, 715, 716 and 717 could be useful for the treatment of dental disease.

Round 2

Reviewer 1 Report

The manuscript has been improved.

Author Response

We thank the editors and the reviewers for their thoughtful and helpful comments.

Reviewer 2 Report

The manuscript is now improved. I appreciate that the authors have restructured the manuscript and provided more methodological detail. My only major concern is that the authors fail to organize the discussion section. Moreover, the manuscript requires minor edits for further clarity (detailed below), typos and grammatical issues.

Some specific comments

Abstract

- lines 14-15: please add “including KHQ 711,712,713,714,715,716,717 and 718 were..” as indicated in ”Biomedicines 814581 supplementary.doc”

- line 33: please change “derivate” in “derivates”

Materials and methods

page 8

-lines 348, 349 and 345: please change “and” to “and”

-line 349: please add “broth” or “agar” after brain heart infusion

-lines 364-365: please mark all microbial names in Italic font

-line 369: please add “in” after “maintened”

Table 1

please exchange KCTC 2640 with KCTC 9146 in the column of description

page 9

-line 373: please change “the “ to “or” between RCM and BHI

-line 377: please change “5” to superscript number

-line 391: please delete “(1.56-12.5µg/mL)” and add “at MIC and ½ MIC concentrations”

-line 393: please change sentence “Growth was evaluated by measuring OD600 using a microplate reader after 0, 6, 12, 18, 24, 30, 36, 42, 48, 54, 60, 66 and 72 h” to “Growth was evaluated by measuring OD600 using a microplate reader at time zero and after 6, 12, 18, 24, 30, 36, 42, 48, 54, 60, 66 and 72 h of incubation”.

Page 10

-line 433: please add “S.sobrinus” after the first KCTC 5809 and “E.faecium” after the second KCTC 5809

-line 434: please add “KACC 13234” after “KACC 16833”

-line 435: please delete “KACC 13234”

Page 12

-line 433: please change “713 (1.2)” to “713 (1.12)”

-line 444: please change “718 (0.98)” to “718 (0.96)”

Page 15

-line 482: please change “727” to “717”

Discussion

In my opinion, this section could be organized as see below:

from the first version of the paper with some modifications

“Dental disease is one of the most prevalent public health concerns. The problems caused by dental caries affect all age groups, and treatment is both expensive and labor-intensive. 19 Dental caries and periodontal diseases are infectious caused by common oral bacteria including Lactobacillus spp., Streptococcus spp. and Actinomyces spp., which usually form plaque biofilms on the tooth surfaces.

E. faecalis is an opportunistic pathogen that is frequently isolated from asymptomatic and persistent endodontic infections, especially from failed root canals undergoing retreatment. 20 E. faecalis is a better survivor than other root canal microbes being able to resist to many antibacterial agents 25.

Among the bacteria included in the present study, viridans streptococci S. sobrinus and S. mutans were the most representative human cariogenic bacteria and are also moderately resistant to antibiotics.26 Therefore, controlling or even reducing the levels of these pathogens is a key step in the prevention and treatment of these diseases. 19,27

to be continued with from lines 495-498 (new version)

For all bacteria strains tested, KHQ 711, 712, 713, 714, 715, 716, 717, and 718 showed the highest stable anti-bacterial activity compared with 1,4-naphthoquinone, Menadione, Juglone, Dichlon, KHQ 701 and KHQ 702 (Table 2A and 2B). A stronger antimicrobial activity was exhibited from a 1,4-naphthoquinone derivate having a pyrimidin-4-one moiety containing both a phenyl group (C2 position) and a benzyl group (N3 position), i.e KHQ 713 compound. This activity was more pronounced for P. gingivalis than for other species such as E. faecalis and A. viscosus.

please explain here why did authors make disk diffusion method and time kill assay only for P.gingivalis

The disk diffusion method has been shown that KHQ 711, 712, 713, 714, 715, 716, 717, and 718 were able to inhibit P. gingivalis strain growth compare to oxytetracycline (Table 3). Moreover,as shown in Figure 2A-H, all these derivates at ½ MIC concentrations, significantly inhibited P.gingivalis until 18 hours of incubation.

please insert text from line 512 to 516
